## RESEARCH ARTICLE

# Tyrosine kinase inhibitors affect sweet taste and dysregulate fate selection of specific taste bud cell subtypes via KIT inhibition

Christina M. Piarowski[1,2], Jennifer K. Scott[1,2], Courtney E. Wilson[1,2], Heber I. Lara[3], Ernesto Salcedo[1], Andrew S. Han[1,4], Elaine T. Lam[5], Peter J. Dempsey[6], Jakob von Moltke[3] and Linda A. Barlow[1,2,*]

## ABSTRACT

Taste dysfunction, or dysgeusia, is a common side effect of many cancer drugs. Dysgeusia is often reported by people treated with anti-angiogenic tyrosine kinase inhibitors (TKIs), which inhibit receptor tyrosine kinases (RTKs). However, the mechanisms by which TKIs cause dysgeusia are not understood, as the role of RTKs in adult taste homeostasis is unknown. Here, we find that treating adult mice with the TKI cabozantinib shifts the fate of differentiating functional taste cell subtypes within taste buds. Through behavioral assays, we find this cell fate shift leads to blunted responses to sweet tastants in cabozantinib-treated mice. Finally, we show that inducible knockout of the RTK KIT, which is inhibited by cabozantinib, phenocopies taste cell fate shifts induced by TKI treatment. Our results establish KIT as a regulator of taste cell homeostasis and suggest that KIT inhibition may underlie TKI-induced dysgeusia in humans.

KEY WORDS: Gustation, Receptor tyrosine kinase inhibitors, Mouse molecular genetic models, Taste dysfunction, Lingual organoids

## INTRODUCTION

The sense of taste, or gustation, plays a central role in how animals experience the world. Taste conveys important nutritional information through five modalities: sweet, bitter, umami, sour and salty (Liman et al., 2014). Taste dysfunction (dysgeusia) is a common side effect of many cancer therapies (Gaillard and Barlow, 2021), which leads to significantly reduced quality of life, including weight loss and depression, as well as worsened long-term outcomes (Epstein and Barasch, 2010; Sánchez-Lara et al., 2010). Improving both quality of life and treatment outcomes therefore necessitates interventions that mitigate or prevent dysgeusia. However, the field lacks important mechanistic understanding of how most cancer therapies affect the taste system (Gaillard and Barlow, 2021).

[1]Department of Cell and Developmental Biology, University of Colorado Anschutz Medical Campus, Aurora, CO 80045, USA. [2]Rocky Mountain Taste and Smell Center, University of Colorado Anschutz Medical Campus, Aurora, CO 80045, USA. [3]Department of Immunology, University of Washington, Seattle, WA 98109, USA. [4]Developing Scholars Internship Program, Graduate Program in Cell Biology, Stem Cells and Development, University of Colorado Anschutz Medical Campus, Aurora, CO 80045, USA. [5]Department of Medical Oncology, University of Colorado Anschutz Medical Campus, Aurora, CO 80045, USA. [6]Department of Pediatrics, Section of Developmental Biology, University of Colorado Anschutz Medical Campus, Aurora, CO 80045, USA.

*Author for correspondence (Linda.barlow@cuanschutz.edu)

E.T.L., 0000-0001-5629-3402; J.v.M., 0000-0003-2894-3353; L.A.B., 0000-0001-7998-2219

One class of targeted cancer therapies that commonly cause dysgeusia are anti-angiogenic tyrosine kinase inhibitors (TKIs). These TKIs reduce tumor vascularization by inhibiting vascular endothelial growth factor receptors (VEGFRs) and platelet-derived growth factor receptor beta (PDGFRβ), both receptor tyrosine kinases (RTKs) (Vigarios et al., 2017; Kiselyov et al., 2007; Raica and Cimpean, 2010). A subset of anti-angiogenic TKIs, including cabozantinib, axitinib and sunitinib, are approved as first- and second-line therapies for metastatic renal cell carcinoma (mRCC) and cause dysgeusia in 10-50% of people with mRCC (Vigarios et al., 2017; Hahn et al., 2019; Roberto et al., 2021; Hamazaki and Uesawa, 2024). However, the intended targets, VEGFRs and PDGFRβ, are rarely found in epithelial cells, as they are expressed primarily in endothelial and stromal cells (Strell et al., 2024; Koch et al., 2011). Further, transcriptome profiling has not detected VEGFR or PDGFRβ mRNAs in murine taste epithelium (Schaum et al., 2018; Sukumaran et al., 2017; Ren et al., 2017; Yamada et al., 2021; Golden et al., 2021; Shechtman et al., 2023; Lee et al., 2017). Cabozantinib, axitinib and sunitinib, as well as TKIs more generally, inhibit many off-target RTKs including RET, MET, KIT and others (Klaeger et al., 2017; Karaman et al., 2008; Davis et al., 2011; Yakes et al., 2011). Many of these off-target RTKs are expressed in taste epithelium (Biggs et al., 2016; Choo and Dando, 2021; Donnelly et al., 2018; Mclaughlin, 2000; Sukumaran et al., 2017; Yamada et al., 2021; Wang et al., 2025), raising the possibility that inhibition of one or more off-target RTKs underlies TKI-induced dysgeusia.

Taste is mediated by taste buds, which are collections of 50-100 specialized, post-mitotic epithelial taste bud cells (TBCs) categorized into three morphological types: type I, type II and type III (Delay et al., 1986; Kinnamon et al., 1993). Type I cells are glial-like support cells, type II cells detect sweet, bitter or umami tastants, while type III cells detect sour and some salty tastants (Liman et al., 2014; Finger and Silver, 2000; Ohtubo and Yoshii, 2011; Ogata and Ohtubo, 2020). Taste buds also house specialized sodium-sensitive cells similar to type II cells in terms of morphology, developmental regulation, and expression of many type II cell markers (Chandrashekar et al., 2010; Nomura et al., 2020; Ohmoto et al., 2020). All TBCs are short-lived, with lifespans ranging from 10 to 40 days, and are steadily generated by proliferating progenitors outside of buds (Perea-Martinez et al., 2013; Beidler and Smallman, 1965; Farbman, 1969) (for reviews, see Barlow, 2015; Piarowski et al., 2025). Despite this continual turnover, the proportions of TBC types are believed to be relatively stable throughout life (Barlow, 2015). Thus, the renewal process, spanning progenitor proliferation to TBC fate specification, differentiation and survival, must be tightly regulated to ensure reliable taste function over time. As TKIs used to treat mRCC are frequently associated with taste dysfunction (Hamazaki and Uesawa, 2024; Vigarios et al., 2017), characterizing how these drugs impact TBC homeostasis may provide cellular and molecular mechanistic insight into how this class of TKIs causes dysgeusia.

Taste buds are housed in specialized gustatory papillae on the tongue. In rodents, fungiform papillae (FFP) are distributed throughout the anterior tongue and contain one taste bud each, while in the posterior tongue, bilateral foliate papillae and a single circumvallate papilla (CVP) at the posterior midline both comprise invaginated epithelial trenches that house hundreds of taste buds (Kinnamon, 1991). Importantly, taste buds in all papillae detect sweet, sour, bitter, etc., although bitter sensitivity is greater in the CVP and sweet sensitivity is greater in anterior FFP (Shingai and Beidler, 1985; Ninomiya et al., 1993). Additionally, specialized sodium-sensing type II-like cells are found only in FFP (Chandrashekar et al., 2010). How TKIs impact taste buds, and whether their impact differs between taste fields, is not known.

Here, we investigated the effects of anti-angiogenic TKIs on taste homeostasis using primary lingual organoids and mouse models. We show that, in the CVP, cabozantinib affects renewal of type II TBC subtypes by dysregulating fate selection of differentiating type II cells such that production of sweet TBCs is diminished and that of bitter/umami cells is increased. Changes in type II TBC composition in CVP taste buds corresponds with blunted sweet taste behavior, thus linking dysregulation of type II TBC fate selection with dysgeusia. Because sweet sensitivity is greater in FFP, we were surprised to find that drug treatment did not block sweet TBC differentiation and instead reduced sodium-sensing type II cells in FFP. Finally, we find that the off-target RTK KIT likely regulates renewal of type II TBC subtypes in both CVP and FFP, as inducible *Kit* knockout in the type II TBC lineage phenocopies TKI treatment. This work provides insight into the regulation of type II TBC subtype fate selection, establishes KIT signaling in this process, and implicates KIT inhibition as a driver of TKI-induced dysgeusia.

## RESULTS

### TKIs selectively affect sweet type II TBCs in lingual organoids

We first used primary lingual organoids to screen the effect of TKIs on taste bud homeostasis, as organoids generated from GFP⁺ progenitors from the CVP epithelium of adult *Lgr5^CreER-eGFP* mice contain all TBC types, non-taste epithelial cells and progenitors (Shechtman et al., 2021; Ren et al., 2014). We assessed whether three TKIs used in treatment of mRCC, cabozantinib, axitinib and sunitinib (Roberto et al., 2021; Hahn et al., 2019), hindered cell proliferation and/or organoid survival during the growth phase of culture (days 2-6) (Fig. S1A). To quantify proliferation, 5-ethynyl-2′-deoxyuridine (EdU) was added to cultures for 30 min prior to harvest; we found that TKI treatment did not alter EdU incorporation by organoids (Fig. S1B,C). We further assessed organoid growth by measuring organoid size at day 6. As a positive control, organoids were treated with paclitaxel, which blocks cell proliferation (Weaver, 2014). These organoids were significantly smaller than negative controls, whereas TKI treatment had little to no impact on organoid growth. Organoids treated with 100 nM axitinib were significantly smaller, but this difference was small compared to the impact of paclitaxel (Fig. S2A). Results with a Cell-Titer Glo® 3D assay, which measures ATP luminescence as a proxy for cell viability, were congruent with organoid size; paclitaxel decreased luminescence compared to controls, whereas TKIs, including 100 nM axitinib, had no impact (Fig. S2B). Lastly, we quantified organoid survival at day 6 and found that 40-60% of isolated *Lgr5*-GFP⁺ progenitors formed organoids regardless of condition (Fig. S2C,D). In sum, our data indicate that cabozantinib, axitinib and sunitinib do not affect taste progenitor proliferation or survival *in vitro*.

Next, to determine whether TKIs affect TBC differentiation, we treated organoids during the differentiation phase of culture (days 6-12) (Fig. 1A). Organoids were harvested for immunofluorescence (IF) for markers of type I (NTPDase2; also known as Entpd2) (Bartel et al., 2006), type II (PLCβ2) (Clapp et al., 2004) and type III (CAR4) (Chandrashekar et al., 2009) TBCs (Table 1). The prevalence of TBC types did not differ in TKI-treated versus control organoids (Fig. 1B,C, Fig. S3A-D). Type II TBCs comprise functional subsets, and those in the CVP that detect bitter or umami tastants express the G protein gustducin (GUST; GNAT3) (Kim et al., 2003) (Table 1). However, GUST⁺ TBC prevalence was also unaffected by TKI treatment (Fig. 1D,E). Thus, TKIs did not impact differentiation of type I, II, III or bitter/umami type II TBCs *in vitro*.

To determine whether TKIs affect other taste epithelial cell populations in organoids, we expanded our panel of markers using RT-qPCR. Expression of *Kcnq1*, expressed by all TBCs (Wang et al., 2009), and *Krt13*, a marker of non-taste CVP epithelium (Iwasaki et al., 2011), was unchanged by drug treatment, suggesting that production of taste versus non-taste lineages was unaffected by TKIs (Fig. S3E,F). Consistent with immunostaining, markers of type I (*Kcnj1*) (Dvoryanchikov et al., 2009), type II (*Plcb2*) and type III (*Pkd2l1*) (Kataoka et al., 2008) TBCs, as well as bitter/umami TBCs (*Gust*) (Table 1) were unaffected by TKIs (Figs 1F,G, Fig. S3G,H). However, we found that markers of the sweet subset of type II TBCs were specifically reduced. Expression of the gene encoding the G protein-coupled receptor TAS1R2, which heterodimerizes with TAS1R3 to mediate sweet transduction (Nelson et al., 2001) (Table 1), was decreased by all three TKIs (Fig. 1H). *Pcdh20*, another sweet cell marker (Hirose et al., 2020) (Table 1), was also reduced significantly by cabozantinib and expression trended downward with sunitinib (*P*=0.07) (Fig. 1I). These data suggest that TKIs selectively affect sweet type II TBCs, but not other cell types, in lingual organoids.

### Cabozantinib changes the cellular composition of taste buds *in vivo*

Since expression of sweet TBC markers was reduced by TKIs *in vitro*, we examined whether TKI treatment specifically affected sweet type II cells in the CVP of adult mice. Cabozantinib was selected because of its long half-life (∼120 h) (Lacy et al., 2017) and because it significantly reduced expression of both sweet TBC markers tested in organoids (Fig. 1). Mice were gavaged with vehicle or cabozantinib for 2 weeks (daily) or 4 weeks (5 days/week) (Fig. 2A,B). Cabozantinib was well tolerated by the mice, whose weights were stable over 4 weeks (Fig. S4A). Taste buds were also grossly unaffected; taste bud number and size did not differ with treatment (Fig. S4B-I). These data suggest that cabozantinib does not affect progenitor cell output or overall renewal of taste buds, consistent with data from organoids (Figs S1, S2).

To investigate cellular changes within taste buds, we first assessed the prevalence of type II (PLCβ2⁺) and type III (SNAP25⁺) (Yang et al., 2000) TBCs (Table 1) in drug-treated versus control mice. As expected from results in organoids, cabozantinib did not alter the number of type II or III cells (Fig. 2C-H). Consistent with reduced *Tas1r2* expression in organoids, however, drug treatment significantly diminished the number of *Tas1r2*⁺ sweet TBCs *in vivo* (Fig. 2I-K). In contrast to organoids, however, where GUST⁺ type II TBCs were unaffected, cabozantinib increased GUST⁺ type II cells in CVP taste buds (Fig. 2L-N). Thus, although the total number of type II cells was unaffected, TKI treatment shifted type II TBC subtype composition; *Tas1r2*⁺ sweet cells were reduced with a concomitant increase in GUST⁺ bitter/umami cells. Importantly, this phenotype

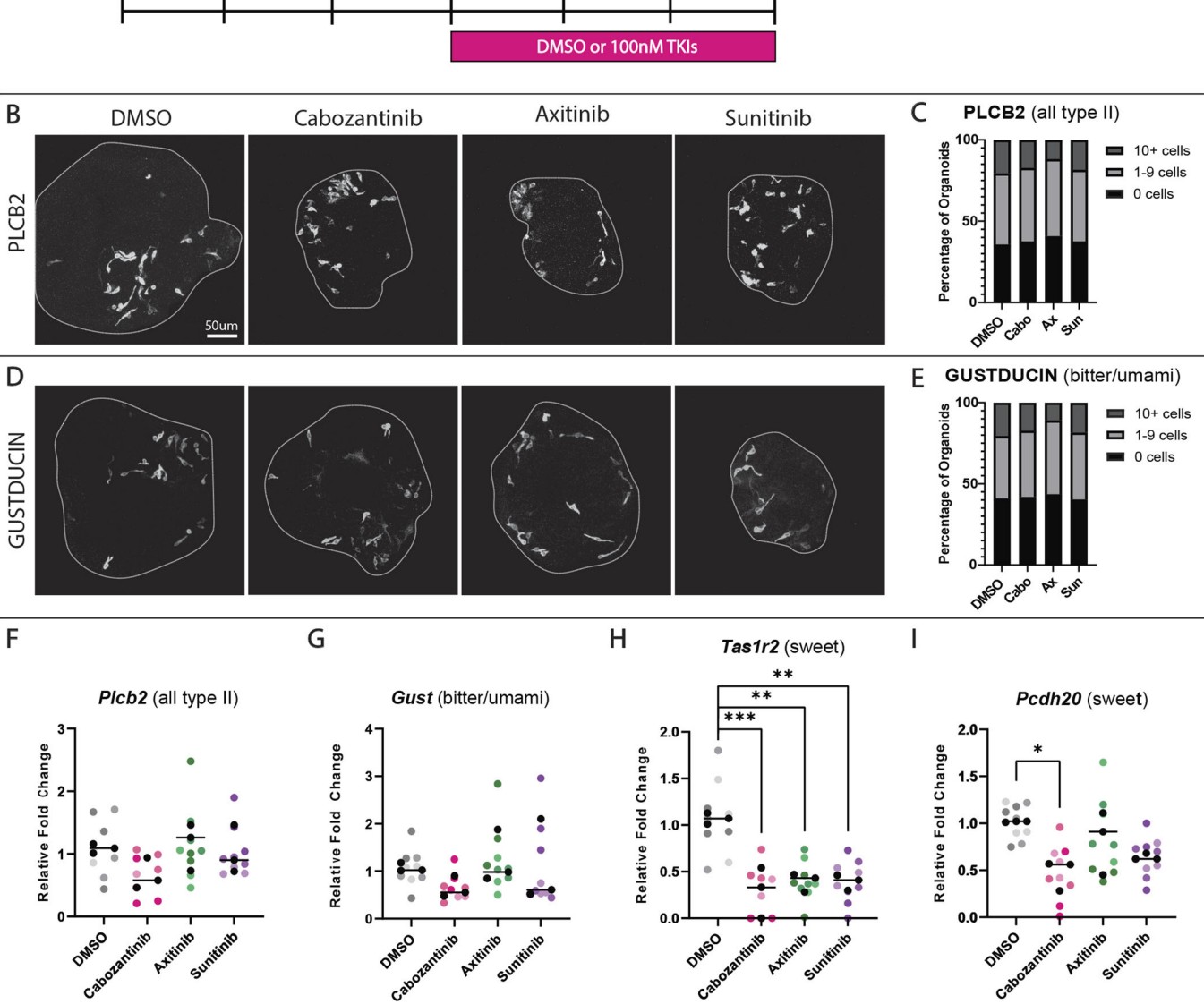

**Fig. 1. TKIs selectively reduce expression of sweet type II TBC markers in organoids.** (A) Experimental design to test cabozantinib (Cabo), axitinib (Ax), and sunitinib (Sun) on TBC differentiation in organoids. (B-E) Compressed confocal z-stacks of control and TKI-treated organoids immunostained for markers of type II TBCs (PLCβ2; B) and bitter/umami type II TBCs (GUST; D). Percentages of organoids containing 0, 1-9 or ≥10 cells immunopositive for PLCβ2 (C) or GUST (E) do not differ with treatment. Total organoids from three independent experiments: DMSO, 112; Cabo, 93; Ax, 110; Sun, 109. (F-I) Relative fold change in type II TBC marker expression by RT-qPCR. Each colored dot represents an individual sample of organoids pooled from three culture wells (over 200 organoids per sample). Within a condition, different shades represent three biological replicates and black dots are averages of each replicate. Ordinary one-way ANOVA with Dunnett's multiple comparisons test (F-H) or Holm–Šídák's multiple comparisons test (I) were used to compare average values (black line) for each marker across conditions (*$P \leq 0.05$, **$P \leq 0.01$, ***$P \leq 0.001$).

was progressively robust with longer drug treatment (Fig. 2), suggesting that the shift in type II subtypes depends on gradual turnover of TBCs.

**Table 1. Comparison of TBC markers in CVP versus FF taste buds**

| TBC type | Circumvallate | Fungiform |
|---|---|---|
| Type I | NTPDASE2, *Kcnj1* | |
| Type II | PLCβ2 | |
| • Bitter/umami | GUST | |
| • Sweet | *Tas1r2*, *Pcdh20* | *Tas1r2*, *Pcdh20*, GUST |
| • Salt | N.A. | GUST negative |
| Type III | SNAP25, CAR4, *Pkd2l1* | |

*In vivo* TKI treatment revealed an increase in GUST[+] bitter/umami cells not seen in organoids. Upon closer inspection, we found that GUST[+] type II cells were over-represented in CVP-derived organoids compared to their proportion in CVP taste buds (80% versus 60%, respectively, of PLCβ2[+] type II cells are GUST[+]) (Fig. S5A,B). Thus, over-representation of GUST[+] TBCs in organoids may have obscured any smaller increases in this cell population caused by drug treatment.

## Cabozantinib does not shift cell type composition via cell death or transdifferentiation

As cabozantinib significantly reduced *Tas1r2*[+] sweet cells and increased GUST[+] bitter/umami cells in the CVP, we reasoned

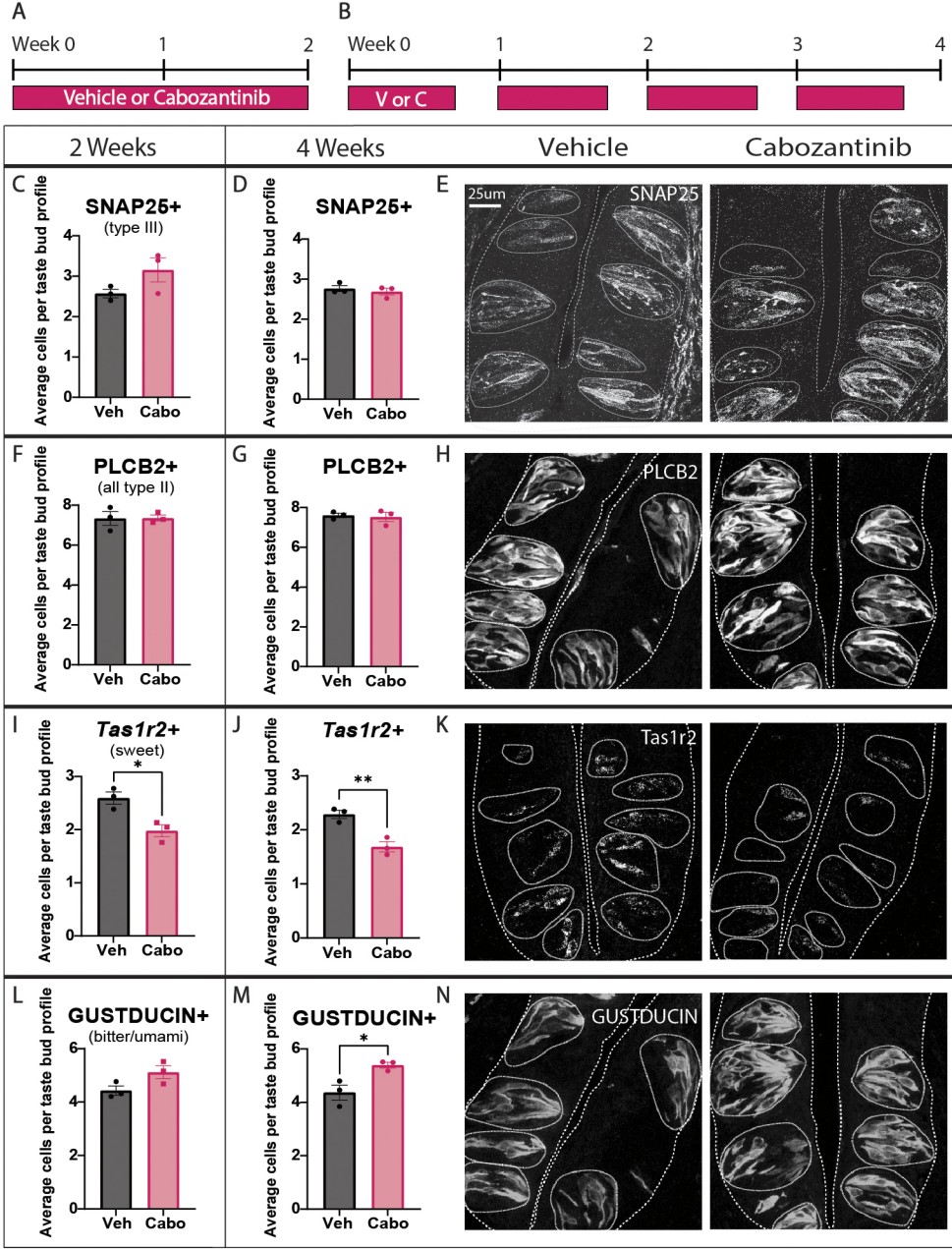

**Fig. 2. Cabozantinib changes the composition of type II TBC functional subtypes in CVP taste buds.** (A,B) Mice were treated daily for 2 weeks (A) or 5 days per week for 4 weeks with drug (Cabo, C) or vehicle (Veh, V) (B). (C-N) Quantification of the average number of stained cells per taste bud profile with corresponding representative images for SNAP25 immunofluorescence (C-E), PLCβ2 immunofluorescence (F-H), *Tas1r2* HCR *in situ* hybridization (I-K) and GUST immunofluorescence (L-N). Representative images are compressed confocal *z*-stacks. Coarse dashed lines delineate basement membrane and apical epithelial surface; fine dashed lines encircle individual taste buds. Note that vehicle-treated images in H and N are also shown in Fig. S5B. Scale bar in E applies to H, K and N. In all histograms, each dot represents the average taste cell tally from each mouse (*N*=3 per condition, ∼80 taste buds/mouse). Unpaired *t*-test. Mean±s.e.m. Each dot is the average value for one mouse. *$P \leq 0.05$, **$P \leq 0.01$.

this shift could be due to: (1) decreased sweet TBC survival, (2) transdifferentiation of sweet cells into bitter/umami cells, or (3) a shift in fate of newly differentiated type II cells. A previous study revealed that the RTK KIT is expressed by TAS1R3⁺ TBCs, which encompass sweet and umami type II TBC subtypes in CVP (Choo and Dando, 2021). More recently, KIT has been identified as a specific marker of sweet cells (Ki et al., 2025). Using hybridization chain reaction (HCR) *in situ* hybridization, we confirmed that *Kit* colocalizes with *Tas1r2* in CVP taste buds, with 98% of *Kit*⁺ cells co-expressing *Tas1r2* (Fig. S6A). Interestingly, *Kit* marks most but not all sweet cells, as 28% of *Tas1r2*⁺ cells do not co-express *Kit* (Fig. S6A,C). Consistent with previous reports (Kim et al., 2003), we found that ∼10% of *Tas1r2*⁺ cells express GUST. However, there was negligible co-expression of KIT and GUST. Thus, Tas1r2⁺/KIT⁺ sweet cells and GUST⁺/KIT⁻ bitter/umami cells are mutually exclusive populations of type II cells in the CVP (Fig. S6B,C). Consistent with co-expression of *Tas1r2* and *Kit*, and

a decrease in *Tas1r2*⁺ cells in drug-treated mice (see Fig. 2), cabozantinib significantly decreased the number of KIT⁺ TBCs per CVP taste bud profile (Fig. S6D-F). These results confirm that *Kit*⁺ sweet cells are affected by drug treatment in CVP taste buds.

To assess the fate of differentiated sweet cells following drug treatment, we induced lineage tracing in *Kit*^CreER/+;*Rosa26*^YFP/YFP mice to label existing *Kit*⁺ sweet cells and then treated animals with vehicle or drug for 2 weeks (Fig. 3A). If cabozantinib decreases sweet cell survival, then we would expect fewer lineage traced *Kit*-YFP⁺ cells with drug treatment. However, *Kit*-YFP⁺ cell number did not differ in vehicle- versus cabozantinib-treated mice, indicating no impact on sweet cell survival (Fig. 3B,C). We next examined type II subtype identity of these *Kit*-YFP⁺ cells to determine whether cabozantinib caused transdifferentiation of differentiated sweet cells, potentially into GUST⁺ bitter/umami cells. Consistent with HCR results, in controls most *Kit*-YFP⁺ cells were PLCβ2⁺ type II cells (Fig. 3D,E). Of these, ∼85% were

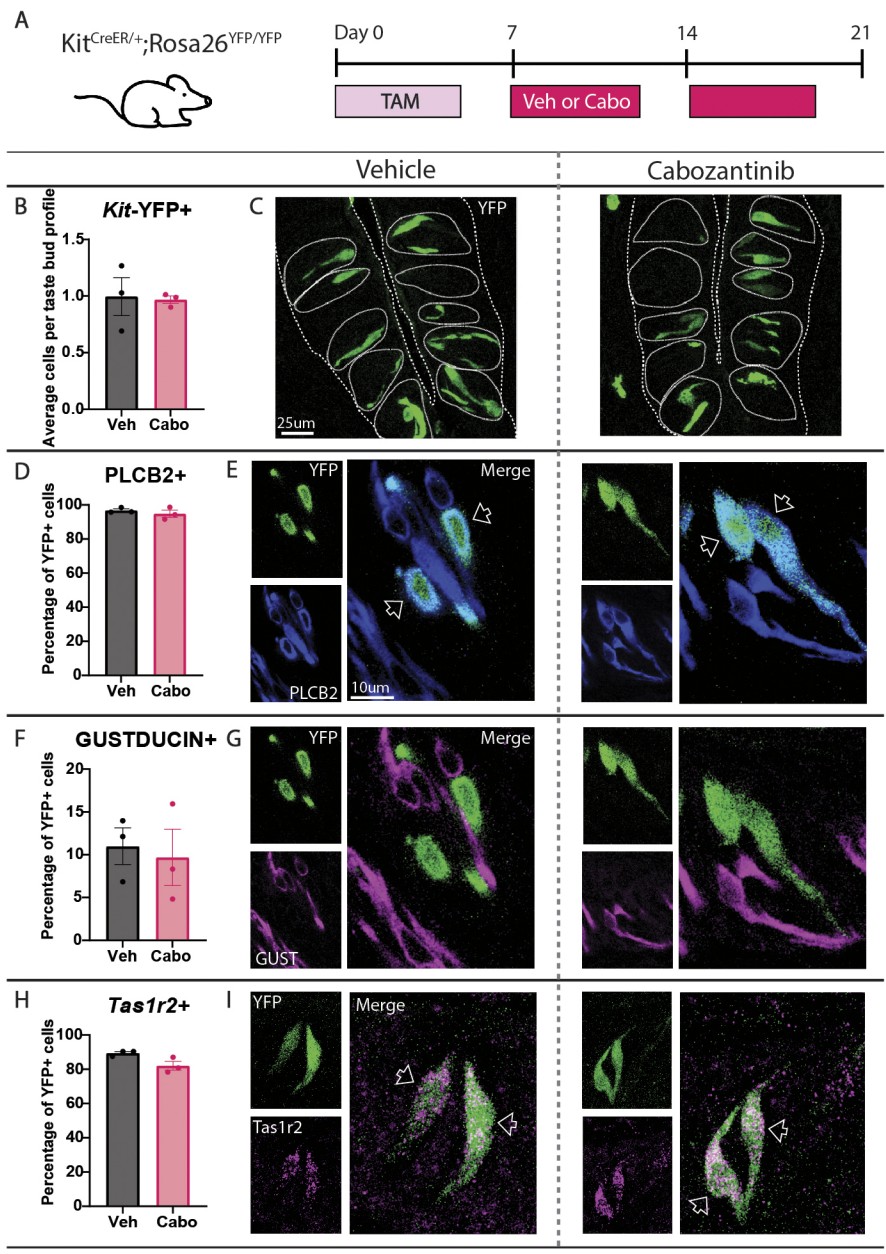

**Fig. 3. Cabozantinib does not impact survival or induce transdifferentiation of sweet type II TBCs in CVP taste buds.** (A) *Kit^CreER/+;Rosa26^YFP/YFP* mice were gavaged with tamoxifen (TAM) daily for 5 days. Following a 3-day chase with daily cage changes, mice were gavaged with vehicle (Veh) or cabozantinib (Cabo) 5 days per week for 14 days. (B) *Kit*-YFP⁺ cells per taste bud profile were unaltered by TKI treatment. (C) Compressed confocal *z*-stacks of *Kit*-YFP⁺ taste cells in CVP taste buds from vehicle- versus cabozantinib-treated mice. Coarse dashed lines delineate basement membrane and apical epithelial surface, fine dashed lines encircle individual taste buds. (D-I) The percentage of *Kit*-YFP⁺ cells (green in all panels) co-expressing PLCβ2 (D,E; blue in E; optical section), GUST (F,G; magenta in G; optical section) or *Tas1r2* (H,I; magenta in I; compressed *z*-stack), is unchanged by drug treatment. Unfilled arrowheads in E and I indicate double-labeled cells. Scale bar in E applies to G and I. In all panels, values were calculated across >200 YFP⁺ cells and >200 taste buds per condition. Unpaired *t*-test performed for all quantifications. Mean±s.e.m. Each dot is the average value for one mouse.

*Tas1r2⁺* sweet cells with very few (∼10%) GUST⁺ bitter/umami cells (Fig. 3F-I). Since KIT and GUST are not co-expressed in differentiated cells (Fig. S6B,C), this suggests that *Kit* is expressed in cells that give rise to both sweet and bitter/umami cells. Regardless, cabozantinib did not change the percentage of *Kit*-YFP⁺ cells expressing *Tas1r2* or GUST (Fig. 3F-I), nor did it change the average number of YFP⁺ cells expressing PLCβ2, GUST or *Tas1r2* (Fig. S7A-C), demonstrating that differentiated sweet cells in the CVP maintain their identity and do not transdifferentiate into bitter/umami type II TBCs with drug treatment.

## Cabozantinib alters the fate of differentiating type II TBC subtypes

We next investigated whether TKI treatment alters the fate of differentiating TBCs. Sonic hedgehog (SHH) marks post-mitotic taste precursor cells that differentiate into each of the TBC types. Precursor cells enter taste buds and express *Shh* transiently (for ∼24 h), then differentiate into TBCs within 3 days (Miura et al.,

2014, 2006). We thus employed *Shh^CreER/+;Rosa26^tdTomato/+* mice to track the fate of new TBCs that differentiate during drug treatment (Fig. 4A).

In the CVP, the number of *Shh*-Tomato⁺ TBCs was unaffected by TKI treatment, indicating that cabozantinib does not impede overall differentiation of TBCs from *Shh⁺* precursor cells (Fig. 4B,C). Cabozantinib also did not affect differentiation of type II TBCs broadly, as the average number of Tomato⁺/PLCβ2⁺ cells was unchanged (Fig. S8A), and ∼30% of *Shh*-Tomato⁺ cells were PLCβ2⁺ regardless of treatment (Fig. 4D,E). Since cabozantinib reduced *Tas1r2⁺* sweet cells with a commensurate increase in GUST⁺ bitter/umami cells (see Fig. 2), we hypothesized that TKI treatment would shift the fate of newly generated type II TBCs from sweet to bitter/umami fate. While the average number of Tomato⁺/GUST⁺ cells did not change (Fig. S8B), cabozantinib significantly increased the percentage of newly differentiated type II cells (Shh-Tomato⁺/PLCβ2⁺) that were GUST⁺ (Fig. 4F,G). Additionally, cabozantinib significantly decreased both the average number of Tomato⁺/*Tas1r2⁺*

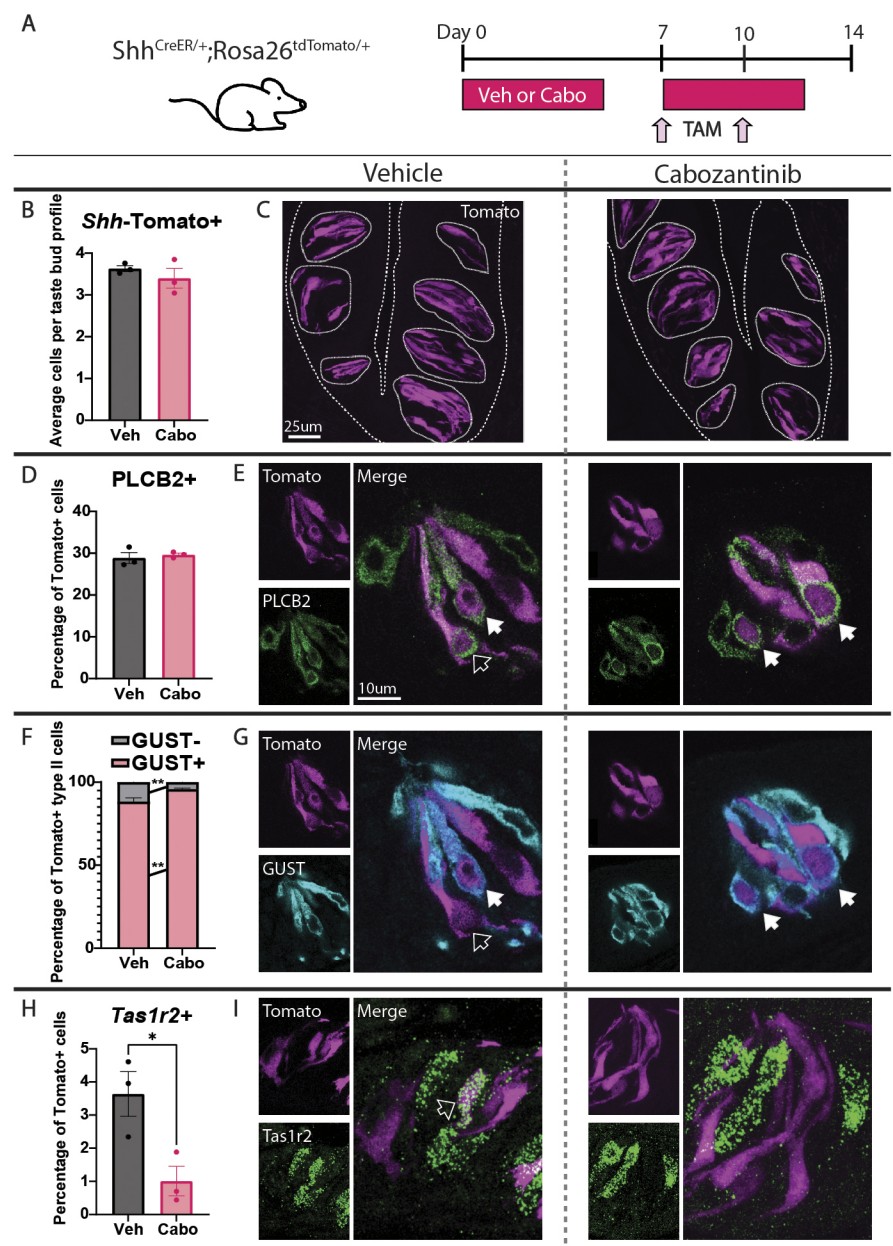

**Fig. 4. Cabozantinib shifts proportions of differentiating type II TBC subtypes in CVP taste buds.** (A) *Shh^{CreER/+};Rosa26^{tdTomato/+}* mice were treated with vehicle (Veh) or cabozantinib (Cabo) 5 days per week for 14 days and dosed with tamoxifen (TAM) on days 7 and 10 to label *Shh^+* taste precursor cells. (B) The number of *Shh*-Tomato^+ cells per taste bud profile was unaltered by cabozantinib. (C) Compressed confocal *z*-stacks of *Shh*-Tomato^+ TBCs in taste buds from vehicle- versus cabozantinib-treated CVPs. Coarse dashed lines delineate basement membrane and apical epithelial surface, fine dashed lines encircle individual taste buds. (D-I) The percentage of *Shh*-Tomato^+ cells (magenta in all panels) co-expressing PLCβ2 did not change (D,E; green in E; optical section). The percentage of Tomato^+/PLCβ2^+ cells expressing GUST significantly increased, while the GUST^− percentage significantly decreased (F,G; cyan in G; optical section). The percentage of *Shh*-Tomato^+ cells expressing *Tas1r2* was significantly decreased (H,I; green in I; compressed *z*-stacks). Unfilled arrowheads in E, G and I indicate double-labeled cells, white arrowheads in E and G indicate triple-labeled cells. Scale bar in E applies to G and I. In all panels, values were calculated across >400 Tomato^+ cells and >200 taste buds per condition. Unpaired *t*-test (B,D,H) and two-way ANOVA with Šidák's multiple comparisons test (F) were performed to compare average values. Mean±s.e.m. Each dot is the average value for one mouse. *$P \leq 0.05$, **$P \leq 0.01$.

cells (Fig. S8C) and the percentage of Tomato^+ cells that were *Tas1r2^+* (Fig. 4H,I). In sum, cabozantinib does not affect differentiation of type II cells in general but rather prevents differentiation of new *Tas1r2^+* sweet cells, instead biasing new type II cells toward a GUST^+ bitter/umami fate. Finally, these results explain why the impact of cabozantinib was greater with prolonged dosing (see Fig. 2), as more TBCs were replaced over time with altered proportions of newly differentiated type II TBC subtypes.

## Cabozantinib impacts taste behavioral preferences of mice
Because sweet TBCs in the CVP were diminished by TKI treatment, we next examined whether sweet taste function was affected. To test this, we performed two behavioral assays: (1) a 48 h two-bottle taste preference test, and (2) a 30 min brief-access taste test. Mice were treated with vehicle or cabozantinib and tested behaviorally during the fourth week (Fig. 5A). Mice prefer sweet tastes, and when presented with a choice will consume/lick more sweet solution than water when sweet is detected above threshold (Reed and Knaapila,

2010). In two-bottle tests, neither vehicle- nor cabozantinib-treated mice preferred sweet solution at low concentrations of the non-nutritive sweetener SC45647 (Tinti and Nofre, 1991). However, control mice strongly preferred 100 μM SC45647, while TKI-treated mice had no preference for sweet taste and drank equal volumes of sweetener and water (Fig. 5B). Importantly, the total volume of solution consumed was not altered by drug treatment, although control mice drank significantly more 100 μM SC45647, likely driven by strong preference for the sweet solution (Fig. S9A).

A brief-access test with Davis Rig lickometers (Smith, 2001) was also used to assess taste behavior. This apparatus allows rapid testing of multiple concentrations of tastant in a 30-min period. Moreover, limited access tests avoid post-ingestive effects that can influence taste behavior (Gaillard and Stratford, 2016). As with two-bottle testing, cabozantinib did not alter important metrics of drinking behavior (Fig. S9B-G). Similarly, control mice in lickometer assays strongly preferred SC45647, while TKI-treated mice showed no preference for the non-nutritive sweetener (Fig. 5C).

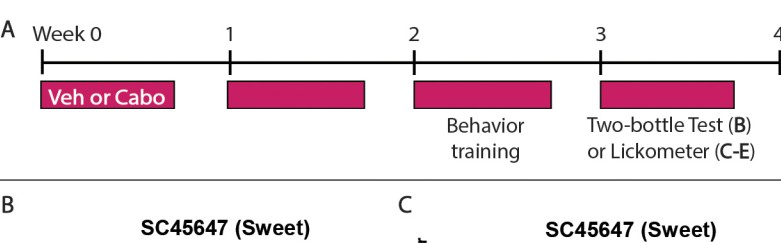

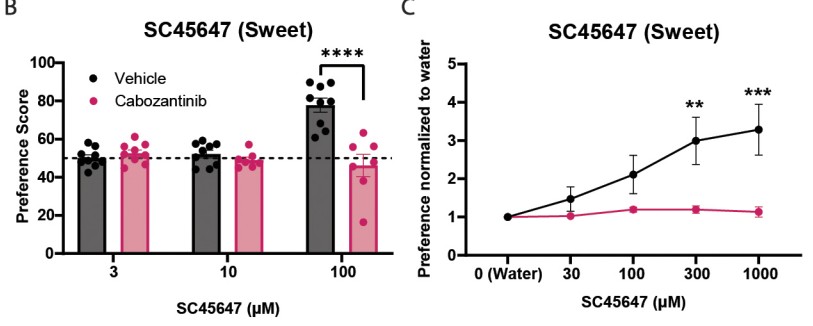

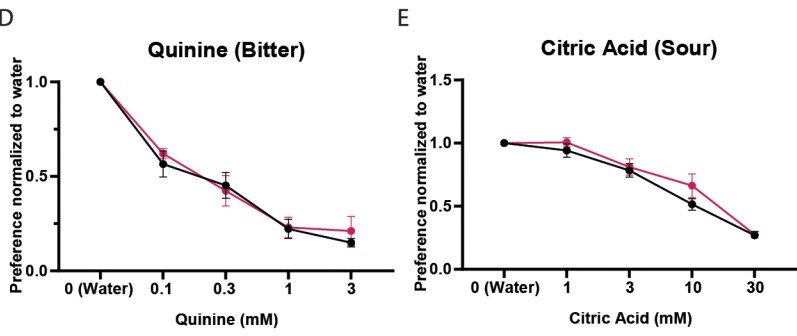

**Fig. 5. Sweet taste preference is diminished by cabozantinib.** (A) Mice treated with vehicle (Veh) or cabozantinib (Cabo) 5 days per week for 4 weeks underwent behavioral assay training in week 3, and taste behavior testing in week 4. (B) In a two-bottle taste test, only control mice preferred sweet solution at high SC45647 concentration ($N$=9 control and 7 TKI mice). Preference score=volume of SC45647 consumed/total volume of water and SC45647 consumed. Dashed line at 50 represents no preference. (C) Brief access lickometer testing shows control mice prefer SC45647 at progressively higher concentrations while TKI-treated mice have no preference for SC45647 over water ($N$=9 control and 9 TKI mice). Lickometer responses for quinine (D) and citric acid (E) were unaltered by TKI treatment ($N$=9 control and 9 TKI mice). Preference=average licks for tastant/average licks for water. All data were analyzed using two-way ANOVA with Šidák's multiple comparisons test. Mean±s.e.m. **$P \leq 0.01$, ***$P \leq 0.001$, ****$P \leq 0.0001$.

Both two-bottle and lickometer tests reveal that cabozantinib decreases behavioral preference for sweet taste, consistent with the histological finding of reduced sweet TBCs in CVP taste buds.

Since cabozantinib increased bitter/umami type II cells, we also tested whether TKI treatment impacted bitter taste function. As bitter is aversive to mice, we hypothesized that the drug-induced increase in GUST$^+$ cells would lower the threshold for avoidance of bitter solution. However, both vehicle- and cabozantinib-treated mice had comparable behavioral avoidance of quinine (Fig. 5D). We also tested sour taste, which is mediated in part by type III TBCs (Liman et al., 2014). Avoidance of citric acid was not affected by drug treatment, consistent with our finding that cabozantinib did not impact type III cells (Fig. 5E). Thus, cabozantinib did not affect taste preference for either bitter or sour but selectively blunted behavioral responses to sweet.

### In anterior FFP, cabozantinib does not affect production of sweet TBCs but instead reduces sodium-sensing TBCs

Although all taste modalities, including sweet, are detected by FFP and CVP taste buds, the FFP taste field is more sensitive to sweet tastants and thus we predicted that sweet type II TBCs in FFP would also be diminished by TKI treatment. In FFP, PLCβ2$^+$ TBCs comprise four functional subtypes: sweet, bitter, umami and sodium sensing. Unlike in the CVP, where $Tas1r2$ and GUST are largely expressed by separate PLCβ2$^+$ cell populations (sweet versus bitter/umami, respectively), almost all $Tas1r2^+$ sweet cells in FFP co-express GUST, as do most bitter and umami type II cells (Kim et al., 2003; Tomonari et al., 2012) (Table 1). In contrast, specialized sodium-sensing TBCs express PLCβ2 and have strong developmental, molecular and morphological similarities with type II cells, but lack GUST expression (Ohmoto et al., 2020; Chandrashekar et al., 2010) (Table 1). Here, we considered all four functional TBC types as

type II subtypes. As cabozantinib did not affect the number of type II or type III cells broadly (Fig. S10A-D), we thus assessed whether type II TBC subtype proportions were altered in FFP after 4 weeks of TKI dosing, and specifically if $Tas1r2^+$/GUST$^+$ sweet TBCs were reduced. Unexpectedly in FFP, neither $Tas1r2^+$/GUST$^+$ sweet cells (Fig. S10I,J) nor GUST$^+$ bitter/umami cells (Fig. S10E,F) were impacted by drug treatment. Instead, cabozantinib treatment resulted in fewer PLCβ2$^+$/GUST$^-$ sodium-sensing cells (Fig. S10G,H).

We also assessed whether cabozantinib induced cell death or transdifferentiation of type II TBCs in FFP (see Fig. 3). As in the CVP, cabozantinib had no impact on TBC survival as the number of $Kit$-YFP$^+$ cells was unchanged (Fig. S11A,B). Interestingly, while $Kit$-YFP primarily labels sweet type II cells in the CVP, $Kit$-YFP lineage traced into type II TBC subtypes with similar proportions in FFP; ~60% of YFP$^+$ cells were GUST$^+$ (30% were $Tas1r2^+$ sweet cells, meaning the other ~30% were likely bitter/umami cells) and ~40% were GUST$^-$ sodium-sensing cells (Fig. S11C-H). Regardless, cabozantinib did not change the percentage of $Kit$-YFP$^+$ cells expressing PLCβ2, GUST or $Tas1r2$ and thus did not induce transdifferentiation in FFP taste buds (Fig. S11C-H).

Finally, we assessed the fate of TBCs newly differentiated from $Shh^+$ precursor cells in FFP taste buds (see Fig. 4A). As in the CVP, drug treatment did not affect differentiation of $Shh$-Tomato$^+$ cells or the percentage of $Shh$-Tomato$^+$ cells differentiating as PLCβ2$^+$ type II TBCs (Fig. S12A-D). Instead, cabozantinib significantly increased the percentage of $Shh$-Tomato$^+$/PLCβ2$^+$ cells that were GUST$^+$ and decreased the percentage that were GUST$^-$ (sodium sensing) (Fig. S12E,F). Since differentiation of $Tas1r2^+$/GUST$^+$ sweet cells was unchanged (Fig. S12G,H), the increase in $Shh$-Tomato$^+$/GUST$^+$ TBCs likely reflects an increase in differentiated bitter/umami type II cells at the expense of GUST$^-$ sodium-sensing TBCs. Thus, cabozantinib alters the fate of newly generated type II

TBCs, albeit affecting different subtypes, in both CVP and FFP taste buds.

## Conditional deletion of *Kit* phenocopies TKI treatment in both CVP and FFP taste buds

Three lines of evidence suggested that TKI inhibition of the off-target RTK KIT may underlie shifts in type II TBC differentiation. In addition to the intended targets VEGFRs and PDGFRβ, KIT is the only RTK inhibited at nanomolar concentrations by all three TKIs tested here (Klaeger et al., 2017; Karaman et al., 2008; Davis et al., 2011; Yakes et al., 2011). Further, mice treated with cabozantinib developed fur depigmentation (Fig. S13), a physiological readout of KIT inhibition in mice and humans (Moss et al., 2003). Finally, *Kit* is expressed in differentiated sweet type II cells in the CVP and lineage traces into all type II cell subtypes in both taste fields, suggesting the type II cell lineage may be directly sensitive to KIT inhibition (see Fig. 3, Figs S6, S11). The transcription factor POU2F3 is expressed in all type II cells and is required for their differentiation (Matsumoto et al., 2011). Thus, we combined the *Pou2F3^CreER* allele (Mcginty et al., 2020) with a novel floxed *Kit* allele (*Kit10*) wherein LoxP sites flank exon 10, which encodes the transmembrane domain of KIT protein (Lara et al., 2026).

To delete *Kit* in the type II TBC lineage, we initially dosed *Pou2F3^CreER/+;Kit^fl/fl* mice with tamoxifen or corn oil every other day for 2 weeks (Fig. S14A). Cre induction abolished KIT expression in CVP epithelium, indicating efficient excision of *Kit* (Fig. S14B,C). As with drug treatment, *Kit* deletion did not affect type II or type III cell number (Fig. S14D-G) and significantly fewer *Tas1r2^+* sweet cells were detected (Fig. S14H,I), although GUST⁺ bitter/umami cell number was unchanged (Fig. S14J,K). We next tested whether longer term *Kit* knockout would fully recapitulate the fate shift in type II TBC subtypes observed after 4 weeks of drug (see Fig. 2). *Pou2F3^CreER/+;Kit^fl/fl* mice were fed tamoxifen chow for 4 weeks (Fig. 6A). As tamoxifen chow can cause weight loss and other side effects (Halpage et al., 2024), control *Kit^fl/fl* mice lacking *Pou2F3^CreER* were also fed tamoxifen chow. Under this protocol, KIT expression was efficiently ablated in CVP epithelium (Fig. 6B,C). Consistent with 4 weeks of cabozantinib treatment, prolonged *Kit* conditional knockout caused a significant decrease in *Tas1r2^+* sweet cells (Fig. 6H,I) with a commensurate, and significant, increase in GUST⁺ bitter/umami type II cells (Fig. 6J,K), while total type II and type III cell numbers were unaltered (Fig. 6D-G). Thus, both cabozantinib and conditional *Kit* knockout induce a shift in type II TBC subtype fate with prolonged treatment (Fig. 6L).

In FFP taste buds, POU2F3 is expressed by all PLCβ2⁺ cells, including sweet, bitter and umami type II cells, as well as by PLCβ2⁺ sodium-sensing taste cells (Ohmoto et al., 2020). As in CVP, *Kit* conditional knockout in *Pou2F3^+* cells efficiently ablated *Kit* in FFP taste buds after 2 weeks of tamoxifen gavage (Fig. S15A,B). As with drug treatment, conditional deletion of *Kit* did not affect the number of type II or type III cells (Fig. S15C,D,G,H), nor were *Tas1r2^+* sweet cells affected (Fig. S15E,F). Rather, consistent with the fate shift induced by cabozantinib in FFP, GUST⁻ sodium cells were decreased and GUST⁺ bitter/umami cells were increased with *Kit* deletion (Fig. S15I-L). Thus, inducible *Kit* knockout phenocopies the effects of cabozantinib treatment in both FFP (Fig. S15M) and CVP, suggesting that TKIs disrupt type II TBC fate selection through KIT inhibition.

## DISCUSSION

Many cancer drugs have the unintended side effect of perturbing taste. Since turnover of TBCs relies upon proliferating progenitors, therapies targeting proliferating cells are especially harmful to the taste system. For example, targeted head and neck irradiation, as well as anti-mitotic chemotherapies, are cytotoxic to taste progenitors and disrupt TBC renewal (Jewkes et al., 2018; Gaillard et al., 2019; Mukherjee et al., 2017; Ren et al., 2023). However, we find that the three anti-angiogenic TKIs tested here do not affect progenitor proliferation or survival when tested in lingual organoid cultures. TBC renewal can also be grossly disrupted by drugs that inhibit taste cell differentiation. Targeted drugs that inhibit SHH signaling, which is required for specification of new TBCs, prevent the differentiation of new taste cells and thus lead to progressive taste bud degeneration as older cells are lost and not replaced (Castillo et al., 2014; Castillo-Azofeifa et al., 2017; Liu et al., 2013; Kumari et al., 2015, 2018; Yang et al., 2015; Lu et al., 2018). TKI treatment, however, did not cause large-scale disruption of TBC differentiation *in vivo* or *in vitro*. Instead, we find that the fate of type II TBC subtypes specifically was dysregulated by anti-angiogenic drugs. In the CVP, cabozantinib decreased sweet type II cells and increased bitter/umami type II cells, while in FFP, sweet cells were not affected. Instead, cabozantinib led to reduced sodium cells with a commensurate increase in bitter/umami cells. Although multiple type II subtypes were altered by drug treatment, surprisingly only sweet taste behavior was impacted. In the CVP and FFP, these shifts in cell fate were recapitulated by deletion of *Kit* in the type II TBC lineage, suggesting that the impact of TKIs on type II TBC fate is due to KIT inhibition.

## Lineage dynamics of taste bud cell renewal

*Shh^+* taste precursor cells differentiate into all TBC types (Miura et al., 2014). However, it is not known whether these cells differentiate directly into TBCs or if further intermediate cell states exist wherein additional lineage decisions are made. *Ascl1* is expressed in type III TBCs and a subset of *Shh^+* precursors, and is required for type III cell production (Seta et al., 2006, 2011). Lineage tracing of *Ascl1^+* cells primarily labels type III cells and a small number of type II cells (Matsuyama et al., 2023; Hsu et al., 2021), suggesting *Shh^+* precursors co-expressing *Ascl1* are capable of producing both cell types but are biased toward type III fate. In contrast, *Pou2f3* is expressed by some precursor cells and all mature type II TBCs and is necessary for type II cell production; *Pou2f3* knockout mice lose type II cells with a corresponding increase in type III cells (Matsumoto et al., 2011). Altogether, these studies suggest a model where *Shh^+* precursor cells become progressively lineage restricted to type III and type II cell fate by *Ascl1* and *Pou2f3*, respectively. However, where in the taste lineage cells acquire type II TBC subtype fates is unknown. Since knocking out *Kit* in *Pou2f3^+* cells, as well as cabozantinib treatment, had no effect on differentiation of type II cells broadly but rather shifted the fate of type II TBC subtypes, we propose that *Shh^+* descendent cells acquire type II subtype fate in POU2F3^+ lineage-restricted type II cell precursors. Since FFP sodium cells also depend on POU2F3 for differentiation (Ohmoto et al., 2020), and since cabozantinib and *Kit* knockout affected the differentiation of sodium cells, our data support published findings that sodium cells and type II cells also arise from a common POU2F3^+ precursor population in FFP taste buds (Ohmoto et al., 2020).

In both CVP and FFP, we find that *Kit* lineage tracing labels all type II TBC subtypes, albeit to different degrees, whereas KIT is not expressed in all differentiated type II cells (Choo and Dando, 2021; Ki et al., 2025) (Fig. 3, Fig. S10).Together, these findings suggest that KIT is expressed in POU2F3^+ type II precursor cells. Additionally, conditional *Kit* knockout in *Pou2f3^+* cells revealed that KIT is required for production of the correct proportions of type II TBC subtypes. Thus, we hypothesize that KIT signaling governs fate selection prior to differentiation of type II TBCs. Interestingly, a

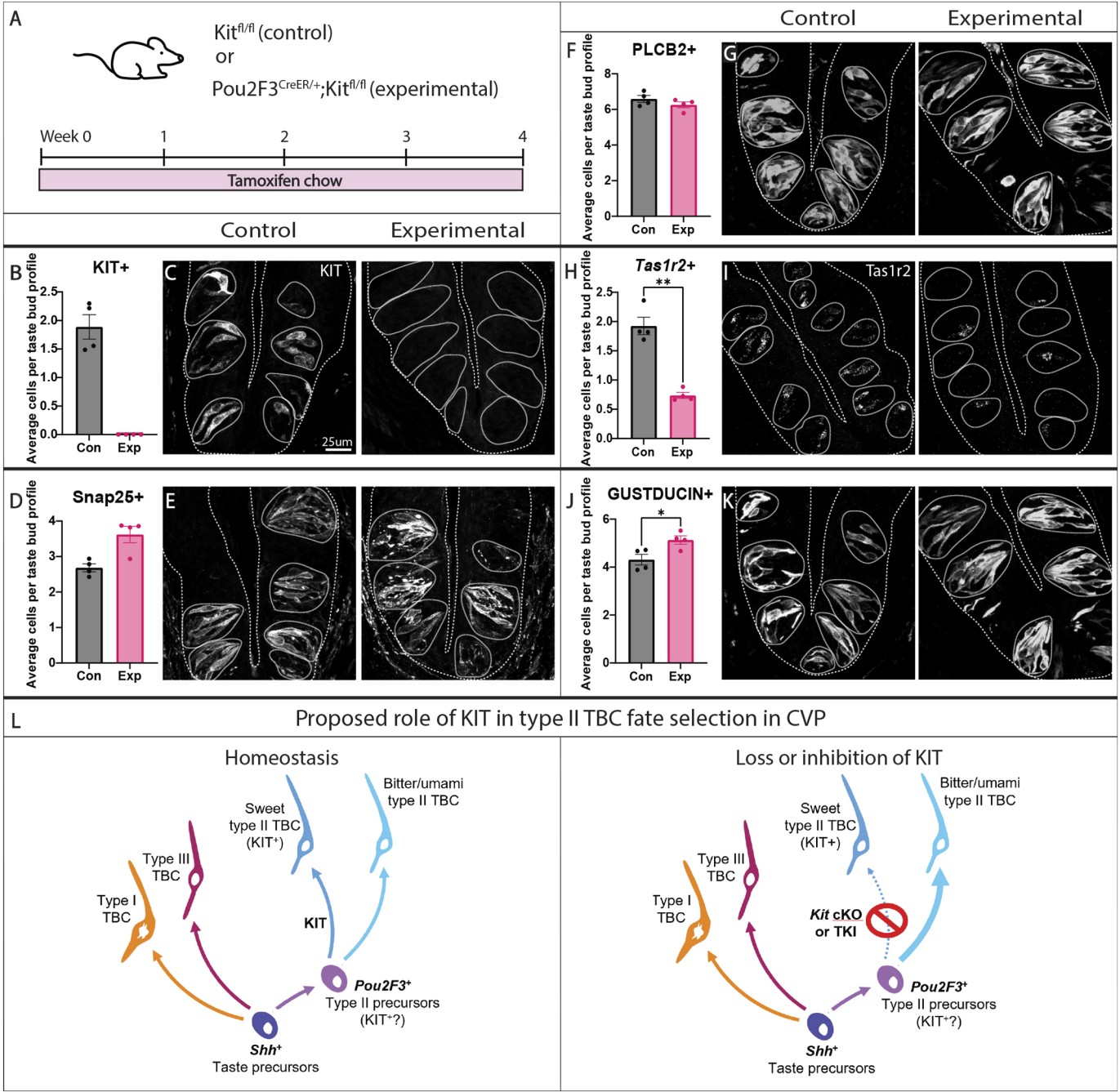

**Fig. 6. Conditional *Kit* knockout in the type II TBC lineage shifts type II TBC subtypes in CVP taste buds.** (A) *Kit* <sup>fl/fl</sup> (control, Con) and *Pou2f3*<sup>CreER/+</sup>; *Kit* <sup>fl/fl</sup> (experimental, Exp) mice were fed tamoxifen chow for 4 weeks. (B-K) Quantification of stained cells per taste bud profile with corresponding representative images for KIT immunofluorescence (B,C), SNAP25 immunofluorescence (D,E), PLCβ2 immunofluorescence (F,G), *Tas1r2* HCR *in situ* hybridization (H,I) and gustducin immunofluorescence (J,K). Representative images are compressed confocal z-stacks. Coarse dashed lines delineate basement membrane and apical epithelial surface, fine dashed lines encircle individual taste buds. Scale bar in C applies to E, G, I and K. In all histograms, each dot represents the average TBC tally from each mouse (*N*=4 per condition, ~80 taste buds/mouse). Unpaired *t*-test performed on all quantifications. Mean±s.e.m. Each dot is the average value for one mouse. *$P \leq 0.05$, **$P \leq 0.01$. (L) Proposed model of *Kit* function in type II TBC lineage, where *Pou2f3*<sup>+</sup> precursor cells gives rise to all type II TBC subtypes, but those destined for a sweet cell fate upregulate *Kit*. Additionally, our pharmacological and genetic data support a model in which KIT function is required for sweet type II cell fate in CVP. Although different in details, we also conjecture a comparable model for FFP taste buds, but instead KIT function in *Pou2f3*<sup>+</sup> type II precursors is required for salt sensing type II cell fate (see Fig. S13). cKO, conditional knockout.

recent report found that the KIT inhibitor imatinib had no effect on normal taste homeostasis (Ki et al., 2025). It is important to note that imatinib has a much shorter half-life than cabozantinib (~20 h versus ~120 h, respectively) (Peng et al., 2005; Lacy et al., 2017), and mice were treated daily for only 12 days. Since KIT inhibition affects type II cells as they gradually turn over, and since sweet cell

reduction was observed after 14 days of either cabozantinib treatment or inducible *Kit* knockout, imatinib may not have been present at high enough levels and/or for long enough for the effects on type II cell fate to become apparent.

KIT regulates the differentiation of multiple cell types in other renewing epithelial systems. For example, in hair follicles, KIT

controls differentiation of melanocytes via the activity of the transcription factor MITF (Liao et al., 2017). KIT also governs differentiation of spermatogonia through transcriptional upregulation of *Stra8*, *Kit*, *Piwil2* and *Oct4* (*Pou5f1*) (Nasimi et al., 2021). Future studies could test whether one or more of these genes functions downstream of KIT in taste homeostasis. Interestingly, KIT signaling is also important in responses to infection and injury. KIT has been proposed to confer resistance to sweet cells following nerve injury (Ki et al., 2025), suggesting that KIT may regulate both homeostatic and repair processes within the taste system. In the intestine, KIT regulates the de-differentiation and plasticity of Paneth cells during inflammation (Schmitt et al., 2018). Additionally, a complementary paper using *Pou2f3*$^{CreER/+}$;*Kit*$^{fl/fl}$ mice identified KIT as a regulator of intestinal tuft cell hyperplasia following helminth infection (Lara et al., 2026). In this context, KIT is required for proliferation and/or differentiation of tuft cells during a type II immune response. Tuft cells, like type II TBCs, require POU2F3 for their generation (Gerbe et al., 2016), suggesting a shared requirement of KIT signaling in regulating the differentiation of POU2F3-dependent cells in multiple epithelial systems.

## Mechanisms of behavioral taste alterations

In rodents, it is well established that FFP taste buds in the anterior tongue are more highly tuned to sweet tastes than are posterior CVP taste buds, while posterior taste buds are more sensitive to bitter stimuli (Ninomiya et al., 1993; Shingai and Beidler, 1985). Thus, we were surprised when cabozantinib had no impact on sweet cell differentiation in FFP despite causing a decline in the behavioral preference for sweet stimuli. Instead, sweet cell loss was evident only in CVP taste buds. These discordant changes in different taste papillae may contribute to TKI-induced dysgeusia, as incongruent input from anterior and posterior taste fields may change how taste stimuli are perceived (as proposed by Gaillard et al., 2017). Sweet taste behavioral responses may therefore depend on balanced sweet cell renewal in FFP and CVP, as normal sweet cell number in FFP did not appear to compensate for reduced sweet TBCs in the CVP. Additionally, it has been proposed that FFP taste buds mediate fine taste discrimination while CVP taste buds regulate a binary accept/reject response once substances reach the posterior oral cavity (Tomchik et al., 2007; Travers et al., 1987; St. John and Spector, 1998). Sweet cells in the CVP could be important to signal tastant acceptance, which may be required to drive appetitive behavior. Thus, a reduction in CVP sweet cells may lower tastant acceptance and decrease the appetitiveness of sweet.

We were also surprised that small reductions in sweet cell number resulted in such large effects on sweet taste behavior. KIT may therefore have additional functions in taste homeostasis. For example, KIT has known roles within the nervous system. *Kit* mutant mice display learning deficits and decreased synaptic potentiation in the hippocampus (Katafuchi et al., 2000). KIT signaling also modulates synaptic activity between cerebellar Purkinje cells expressing KIT ligand and neighboring KIT$^+$ interneurons (Zaman et al., 2024). KIT may therefore function to modulate signaling between KIT$^+$ taste cells and gustatory neurons. In support of this, transcriptome profiling of gustatory neurons in the geniculate ganglion, which innervate FFP taste buds, reveals expression of both *Kit* and KIT ligand (*Kitl*), suggesting that KIT function may be important for proper transmission of taste information to the brain (Zhang et al., 2019; Anderson and Larson, 2020 preprint). Additionally, recent findings suggest that KIT function confers resistance to sweet cell loss following nerve injury, and that these residual sweet cells may promote re-innervation of taste buds by regenerating nerve fibers (Ki

et al., 2025). Together, these studies suggest that the loss of sweet preference in cabozantinib-treated mice could be due to effects on sweet-sensing gustatory neurons in addition to the effects on type II TBC differentiation.

By contrast, behavioral preferences for bitter and sour tastants were unaffected by TKI treatment. Since GUST expression is associated with bitter cells in both CVP and FFP taste buds (Kim et al., 2003; Tomonari et al., 2012), we expected the increase in GUST$^+$ type II cells to cause hypersensitivity to bitter tastants. However, behavioral sensitivity to bitter compounds may already be maximal. Many plants, fungi and animals produce toxic or poisonous substances to deter predation; these are almost always bitter, and thus the taste system has evolved to detect and allow avoidance of bitter substances at low concentrations (Reed and Knaapila, 2010). If bitter taste sensitivity is already high, small increases in bitter cells would have little functional consequence. Overall, our data suggest that loss of sweet taste function is a primary cause of TKI-induced dysgeusia in mice. Intriguingly, sweet taste was reportedly the most commonly affected taste modality in people treated with TKIs for gastrointestinal stromal tumors, while salt taste was the second most commonly affected taste modality (Van Elst et al., 2022).

## Limitations and future directions

Although TKI treatment decreased sodium-sensing type II cells in FFP taste buds, we did not test salt behavior in TKI-treated mice. FFP sodium TBCs mediate detection of low, appetitive concentrations of salt (Ohmoto et al., 2020). Practically, motivating mice to prefer such low concentrations requires dietary salt depletion that may not be well tolerated by TKI-treated mice (Chandrashekar et al., 2010). Additionally, we did not measure the effects of TKIs or conditional *Kit* knockout on umami taste. In large part, this is because the identity of the taste cell type(s) responsible for umami sensing is unsettled. While some work suggests a model in which umami is transduced by a single type II subtype (Zhao et al., 2003; Zhang et al., 2003), other published data suggest that umami detection is more widespread across type II TBCs. For example, in both the CVP and FFP, many *Tas1r2*$^+$ sweet cells, as well as many GUST$^+$ bitter cells, express *Tas1r1*, which with TAS1R3 forms the heterodimeric umami receptor (Kim et al., 2003; Nelson et al., 2002). Further, in a recent report in fungiform taste buds, a significant number of type II cells express both *Tas1r1* and *Tas1r2*, in addition to *Tas1r3*, and importantly, individual type II cells responded to both sweet and umami stimuli (Lee et al., 2025). Thus, assessing the role of KIT signaling in the differentiation of umami-sensing type II cells awaits clarification of the cell type(s) responsible.

Lastly, while the taste system of mice is broadly similar to that of humans (Tizzano et al., 2015), we can only speculate that humans experience similar cell type shifts and changes in taste preferences. This limitation could be addressed in the future by collecting taste bud biopsies from TKI-treated individuals to investigate whether cellular changes are similar to those observed in mice, and by performing psychophysical taste testing for individual tastants (Doty, 2019) in people experiencing dysgeusia.

In closing, given the importance of sweet cells in the maintenance of sweet taste preference, and given the role of KIT in regulating sweet cell renewal in CVP taste buds, we propose that KIT inhibition contributes to TKI-induced dysgeusia. In addition to the three TKIs tested here, KIT is an off-target RTK of seven anti-angiogenic TKIs currently used in clinical practice (Klaeger et al., 2017; Chang et al., 2022; Sánchez-Gastaldo et al., 2017; Yakes et al., 2011). Thus, KIT inhibition may underlie dysgeusia in this large class of targeted cancer therapies. For drugs used to treat

mRCC, KIT is not an intended target, as therapeutic efficacy relies primarily on VEGFR and PDGFRβ inhibition (Sánchez-Gastaldo et al., 2017). Developing alternatives that maintain inhibition of these targets without inhibiting KIT could alleviate dysgeusia while maintaining efficacy in treating mRCC. There are, however, also cancers in which oncogenic KIT mutations drive tumorigenesis, such as gastrointestinal stromal tumors, melanomas, small cell lung cancer and many leukemias (Lennartsson and Rönnstrand, 2012). Unfortunately, treating these cancers requires KIT inhibition and therefore dysgeusia is likely unavoidable. Developing targeted therapies that specifically inhibit mutated KIT without inhibiting wild-type KIT may mitigate dysgeusia in these cases. By establishing KIT as a regulator of type II TBC subtype renewal and showing that *Kit* knockout underpins the response to TKI treatment, we have provided a promising candidate for future mitigation strategies.

## MATERIALS AND METHODS
### Animals
Commercially available mice were obtained from The Jackson Laboratory (*Lgr5^EGFP-IRES-CreERT2*, 008875; *Shh^CreERT2*, 005623; *Rosa26^TdTomato*, 007909; *Rosa26^YFP*, 006148; *Pou2f3^CreERT2-IRES-eGFP*, 037511). *Kit^CreER/+*; *Rosa26^YFP/YFP* mice were a gift from Stephen W. Santoro (University of Colorado Anschutz Medical Campus, Aurora CO, USA; originally reported by Klein et al., 2013). *Pou2f3^CreERT2-IRES-eGFP/+;Kit^fl/fl* mice were a gift from Jakob von Moltke (University of Washington, Seattle, WA, USA; Lara et al., 2026). Mice were maintained in an AAALAC-accredited facility in compliance with the Guide for Care and Use of Laboratory Animals, Animal Welfare Act, and Public Health Service Policy. Male and female adult mice between 8 and 20 weeks were used. Procedures were approved by the Institutional Animal Care and Use Committee at the University of Colorado Anschutz Medical Campus.

### Administration of drugs
#### Cabozantinib malate
Cabozantinib (XL 184, SelleckChem) was suspended in HPLC water at 6 mg/ml with 5 µl/ml 1 N HCl (Fisher Chemical), sonicated for 10 min, heated at 37°C for 15 min with vortexing, and sonicated for another 10 min. Adult mice on a mixed background were individually housed and dosed with 60 mg/kg cabozantinib or vehicle by oral gavage. Mice were weighed daily to monitor health.

#### Tamoxifen gavage
Tamoxifen (Sigma-Aldrich) was suspended in corn oil+10% ethanol at 10 mg/ml. Adult mice were individually housed and gavaged with 50 mg/kg (*Shh^CreER/+;Rosa26tdTomato/+*) or 100 mg/kg (*Kit^CreER/+;Rosa26^YFP/YFP* and *Pou2f3^CreER/+;Kit^fl/fl*). Cages were changed daily for 7 days after final tamoxifen dose or until mice were harvested, whichever came first.

#### Tamoxifen chow
Control (*Kit^fl/fl*) and experimental (*Pou2f3^CreER/+;Kit^fl/fl*) mice were individually housed and fed tamoxifen chow (Inotiv, 500) *ad libitum* for 4 weeks. Mice were weighed daily for the first 10 days to monitor weight loss, then weighed three times per week for the remainder of the study once their weights recovered and stabilized.

### Behavioral assays
#### Two-bottle taste preference test
Preference for SC45647 was tested in a 48 h two-bottle behavioral assay (Gaillard and Stratford, 2016). Mice were individually housed in regular ventilated cages with *ad libitum* chow. Graduated bottles were created from cut plastic serological pipettes fitted with a sipper tube on one end and a rubber stopper on the other. Mice were trained for 4 days; first with deionized water in one of two bottles, and positions switched after 24 h. In the second 2 days, mice had access to water in both bottles. To test sweet taste preference, mice had one bottle of water and one bottle of SC45647 (3 µM in deionized water) for 48 h; the volume consumed was measured at

24 h, bottles refilled as necessary, and left-right position switched to control for side preference. Sweet preference ratio was calculated by dividing the volume of SC45647 consumed by the sum of the volume of water and SC45647 consumed over 48 h. This was repeated for 10 µM and 100 µM SC45647.

#### Brief access test (lickometer)
Three Davis Rig MS-160 lickometers (DiLog Instruments, Inc.) were used to test taste preference of vehicle- and cabozantinib-treated mice. Detailed protocol and operating procedures have been described fully (Gaillard and Stratford, 2016). During the last 2 weeks of drug or vehicle treatment, mice were trained and tested in one of three lickometers randomly assigned each day. Chamber areas were reduced by half to reduce exploration and distraction from surroundings. Training and test sessions were limited to 30 min. During brief-access testing, a shutter opens to give mice access to a bottle of tastant for 5 s (counted from first lick), the shutter closes for 7.5 s, and the next bottle is moved into position by an automated mobile rack for the next 5 s trial. Mice were deprived of water for 23.5 h prior to training or testing to motivate drinking. Mice were trained to drink water in the lickometer for four consecutive days: For 2 days, the shutter was open and water available for 30 min; then on days 3 and 4, two bottles of water were alternately presented for 5 s each for 30 min. After 2 days of recovery, mice had a final day of training with two alternating bottles of water. Once trained, mice were tested for behavioral taste preference for 4 days. For each cabozantinib versus vehicle experiment, only two of a total of three tastants were assayed, each for 2 days. These included: sweet – SC45647 (30, 100, 300 and 1000 µM); bitter – quinine (0.1, 0.3, 1 and 3 mM); and sour – citric acid (1, 3, 10 and 30 mM). All solutions were made in deionized water. Each test session consisted of up to eight trial blocks (each block comprised five presentations in random order: water and four concentrations of one tastant). For sweet SC45647, the first block was not included in our analysis as a thirst-induced high motivational state during the first block may drive maximal licking for all solutions (Gaillard and Stratford, 2016). Mouse thirst for all tastants was calculated from the total licks across the first block averaged across the two testing days. Results of behavior tests are expressed as the lick ratio (average licks to tastant/average licks to water).

### Tissue preparation
Mice were euthanized by $CO_2$ asphyxiation. Tongues were dissected from the lower jaw, incubated for 3 h at 4°C in 4% paraformaldehyde (PFA) in 0.1 M phosphate buffer (PB), and placed in sucrose (20% in 0.1 M PB) overnight at 4°C. Samples were embedded in OCT Compound (Tissue-Tek), 12 µm cryosections collected on SuperFrost Plus slides (Thermo Fisher Scientific) and stored at −80°C. For the CVP, six sets of nine sections were collected, while for the FFP, six sets of 16 sections were collected.

### Immunofluorescence
#### Tissue sections
Sections were washed in 0.1 M PBS, incubated in blocking solution [BS: 5% normal goat or donkey serum, 1% bovine serum albumin (BSA), 0.3% Triton X-100 in 0.1 M PBS at pH 7.3] for 1.5 h room temperature (RT) followed by primary antibodies (Table S2) diluted in BS overnight at 4°C. Sections were rinsed in PBS with 0.1% Triton X-100, incubated for 1 h at RT in secondary antibodies in BS, followed by DAPI nuclear counterstaining (Invitrogen; 1:10,000) and washes with 0.1 M PB. Slides were cover-slipped with ProLong Gold mountant (Thermo Fisher Scientific).

#### Organoids
Organoids were generated from taste progenitor cells obtained by FACS of dissociated CVP epithelia from *Lgr5^EGFP-IRES-CreERT2* mice following our published protocol (Shechtman et al., 2021). To harvest, organoids were incubated in Cell Recovery Solution (Corning) at 4°C, washed in 0.1 M PBS, fixed in 4% PFA and stored at 4°C in PBS with 1% BSA. For immunofluorescence, organoids were incubated in BS (2 h), then with primary antibodies (Table S2) in BS for three nights at 4°C, washed in PBS with 0.2% Triton X-100 and incubated with secondary antibodies in BS overnight at 4°C. Organoid nuclei were counterstained with DAPI, washed

with 0.1 M PB and mounted on SuperFrost Plus slides in Fluoromount (Southern Biotech).

## HCR RNA-fluorescence *in situ* hybridization

Molecular Instruments designed and produced probes against *Tas1r2* (NM_031873.1), *Kit* (NM_001122733.1) and *Trpm5* (NM_020277.2). Methods were adapted from the manufacturer's protocol. Fixed frozen sections were incubated in 4% PFA for 10 min at RT, washed with 0.1× PBS and incubated in 2 µg/ml Proteinase K for 2 min (CVP) or 5 min (FFP) at RT. Sections were incubated in triethanolamine solution, 12 M HCl and acetic anhydride in DEPC-treated water for 10 min at RT, washed with 0.1× PBS, incubated in hybridization buffer for 45 min at 37°C, and then with 1.2 pmol (CVP) or 1.5 pmol (FFP) probe in hybridization buffer overnight at 37°C. Sections were washed with 75% wash buffer/25% 5× SSCT (20× SSC and 10% Tween20, ultrapure water), 50% wash buffer/50% 5× SSCT, 25% wash buffer/75% 5× SSCT, and 100% 5× SSCT, each for 20 min at 37°C, then with 100% 5× SSCT for 20 min at RT. Sections were incubated in amplification buffer for 1 h, then in denatured hairpin solution (6 pmol hairpin 1 and 2 in amplification buffer) overnight. Slides were washed with 5× SSCT, nuclei counterstained with DAPI, washed with 0.1 M PB and cover-slipped with ProLong Gold.

## Organoid derivation

Organoids were derived according to our published protocol (Shechtman et al., 2021). For each experiment, organoids were acutely derived from CVP epithelium from three to six *Lgr5^EGFP-IRES-CreERT2* mice aged 8-20 weeks. Briefly, collagenase (2 mg/ml) and dispase (5 mg/ml) in PBS were injected around the CVP, then the epithelium peeled and dissociated for 45 min in collagenase (2 mg/ml), dispase (5 mg/ml) and elastase (2 mg/ml) at 37°C. Cells were centrifuged (320 $g$ at 4°C), the pellet resuspended in FACS buffer [1 mM EDTA, 25 mM HEPES (pH 7.0), 1% fetal bovine serum, 1× $Ca^{2+}$/$Mg^{2+}$-free dPBS], passed through a 30 µm cell strainer, stained with DAPI (Thermo Fisher Scientific) and subjected to FACS on a MoFlo XDP100 (Cytomation). Debris and doublets were gated out via side-scatter (FSC and SSC-width, respectively), live (DAPI^neg) cells were enriched, and the gating of GFP^+ signal was determined relative to background fluorescence of epithelial cells from a wild-type control mouse processed in parallel (see Shechtman et al., 2021 for gating strategy). Only *Lgr5*-GFP^+ cells were collected. Sorted GFP^+ cells were plated in 48-well plates at 200 cells/well in 15 µl matrix gel (Cultrex RGF BME) and grown in WENR+AS media to support growth (days 0-6) and WENR media to promote TBC differentiation (days 6-12). WENR media consisted of: 50% WRN (WNT/RSPO/Noggin)-conditioned media (Miyoshi and Stappenbeck, 2013) plus 1× GlutaMAX (Gibco), 1× HEPES (Gibco), 1× penicillin-streptomycin (Gibco), 1× B27 supplement (Gibco), 1× gentamicin (Gibco), 1× primocin (InvivoGen), 25 ng/ml murine noggin (Peprotech), 50 ng/ml murine EGF (Peprotech), 1 mM nicotinamide (Sigma-Aldrich) and 1 mM N-acetyl-L-cysteine (Sigma-Aldrich). WENR+AS media consisted of: WENR plus 500 nM A83-01 (Sigma-Aldrich) and 0.4 µM SB202190 (R&D Systems). Y27632 (10 µm; Stemgent) was added on days 0-2 to promote cell survival.

## Drug treatment

Cabozantinib (SelleckChem, S4001), axitinib (SelleckChem, S1005), sunitinib (SelleckChem, S7781) and paclitaxel (SelleckChem, S1150) were dissolved in DMSO to a stock concentration of 50 µM, and diluted in organoid media (WENR+AS or WENR depending on the experiment) to a final concentration of 50 nM or 100 nM. For controls, the same volume of DMSO used for 100 nM drug was added to organoid media. Medium was changed every 2 days.

## Quantitative RT-PCR

Organoids were harvested as described (Shechtman et al., 2021). Briefly, plates were placed on ice for 30 min and organoids freed from matrix gel by scratching with a pipet tip. Organoid samples were centrifuged (300 $g$ for 5 min) and resuspended in remaining media then centrifuged again at 300 $g$ for 5 min. RNA was extracted using a RNeasy Micro Kit (QIAGEN), quantified using a NanoDrop spectrophotometer (Thermo Fisher Scientific) and reverse transcribed with an iScript cDNA synthesis kit (Bio-Rad). Power

SYBR Green PCR Master Mix (Applied Biosystems) was used for qPCR reactions on a StepOne Plus Real-Time PCR System (Applied Biosystems, Life Technologies). Relative gene expression was assessed using the ΔΔCT method (Livak and Schmittgen, 2001), with *Rpl19* as the housekeeping gene. Primers are listed in Table S1. Organoids from three wells of a 48-well plate were pooled per RNA sample and three samples were collected per experiment for RT-qPCR.

## EdU incubation, detection and quantification

Organoids were grown in 48-well plates as described above and on day 6 EdU (4 mM stock in 0.9% NaCl) was added to WENR+AS media at a final concentration of 10 µM for 30 min. Cultures were placed in fresh WENR+AS on ice and organoids harvested for immunofluorescence as above. A Click-it® EdU Alexa Fluor® 647 Imaging Kit (Invitrogen) was used to detect EdU signal using methods adapted from the manufacturer's protocol. Organoids were washed in PBS with 0.2% Triton X-100, washed in PBS with 0.1% BSA, then incubated in Click-it® Plus reaction cocktail for 3 h at RT. Organoids were washed with PBS, nuclei counterstained with DAPI, washed with 0.1 M PB and mounted on SuperFrost Plus slides in Fluoromount (Southern Biotech). Using a custom app – OrganoidAnalyzer in MATLAB 2024a (MathWorks; https://github.com/salcedoe/OrganoidAnalysis) – we determined the ratio of EdU^+ pixels to total DAPI^+ pixels in each organoid.

This app allows the user to select and open a confocal image stack and visually inspect each stack by channel and optical section. The app also provides the controls to process and segment the stack, and capture and collate the data. The app uses the Bio-Formats MATLAB toolbox (Linkert et al., 2010) to open each image stack. Once a confocal z-stack was opened, each channel in the stack was processed separately. Image stacks were normalized to the maximum intensity found in the channel (using the 'mat2gray' function). Stacks were then filtered using a median filter with the default neighborhood settings (medfilt3) and a Gaussian filter using a 5×5×5 filter size (imgaussfilt3). To increase contrast and enhance separation between nuclei in an organoid, an image stack marker was created by applying an erosion morphological operation on the filtered stack (imerode) using a sphere-shaped structuring element with a radius of 3 voxels (strel). A morphological reconstruction of the filtered volume was then performed using the image stack marker created in the previous step (imreconstruct). To identify labeled nuclei, image stacks were segmented into labeled nuclei and background using a modified version of Otsu's method for thresholding. Specifically, a threshold value was calculated for each channel using 'graythresh' and this value was multiplied by a set factor (i.e. 0.5 for the DAPI channel and 0.2 for the EdU channel). The same factors were used for all image stacks to ensure consistent segmentation. To create the binarized segmentation image stack, pixels with intensity values greater than the threshold value were labeled with a logical one to signify labeled cells, while pixels with intensity values below the threshold value were labeled with logical zeros to signify background. Two separate segmentation stacks were created: one for the EdU-labeled cells and one for the DAPI-labeled cells. Each segmentation stack was visually inspected and compared to the original image stack to ensure that pixels designated as signal were in fact signal and not noise. The ratio of EdU^+ nuclei to total DAPI^+ nuclei was calculated by dividing the total number of logical ones in the EdU binarized image stack by the total number of logical ones in the DAPI binarized image stack. As such, the total volume of EdU^+ nuclei was in effect divided by the total volume of all DAPI^+ nuclei to calculate the ratio. This was necessary as our technique was unable to separate all individual nuclei in an organoid, especially the DAPI^+ nuclei which are tightly clustered together.

## Cell-Titer Glo® 3D

Organoids were derived and cultured as above but in opaque-walled 96-well plates at 100 cells per well in 3 µl matrix gel. On day 6, plates were equilibrated to RT for 30 min, after which an equal volume of Cell-Titer Glo® 3D reagent (Promega) was added to cultures (150 µl WENR+AS with drugs or DMSO+150 µl reagent in each well). Plates were mixed on an automatic plate shaker for 5 min to induce cell lysis. Luminescence was recorded after 25 min using a Synergy H1 microplate reader (BioTek). The average luminescence across six wells was calculated for each condition in each biological replicate.

## Image acquisition and analysis

### Immunofluorescence

Tissue sections and organoids were imaged with a Leica TCS SP8 laser-scanning confocal microscope with LAS X software. *z*-stacks of optical sections of CVP and FFP sections were acquired at 0.75 µm thickness, and organoids at 2 µm optical section thickness. For image acquisition, analysis and cell counts using LAS X Office software, investigators were unaware of the condition. For CVP, all taste buds in five tissue sections were counted. For FFP, all taste buds in 16 sections from the tongue tip were counted. Criteria for immunolabeled cell counts were: (1) cell is immunomarker positive; and (2) has a DAPI$^+$ nucleus. Taste buds were identified during cell counting using either PLCβ2 staining (co-staining with GUST and YFP), KRT8 staining (co-staining with SNAP25 and KIT), or *Trpm5* HCR *in situ* hybridization (with *Tas1r2* probe).

### Taste bud area

Maximum *z*-stack projections of KRT8 immunofluorescence tissue sections were generated and imported into ImageJ software. A scale bar was used to set pixels/µm scale. Individual taste buds were outlined using the 'Freehand Selection' tool to obtain the area of each taste bud profile.

### Statistical analyses

In experiments with two conditions, normally distributed data were analyzed by unpaired *t*-test, and Mann–Whitney *U* tests were used when data were not normally distributed. In experiments with more than two conditions, normally distributed data were analyzed by ordinary one- or two-way ANOVA with either Šidák's or Tukey's multiple comparisons post-hoc test. A Kruskal–Wallis comparison test was used when data were not normally distributed. All statistical analyses employed GraphPad Prism software. Data plotted throughout are depicted as mean ±s.e.m. and significance was taken as $P<0.05$ with a confidence interval of 95%.

### Acknowledgements

We thank the University of Colorado Organoid and Tissue Modeling Shared Resource (OTMSR) for providing WRN conditioned media and technical assistance for organoid experiments. We thank Dmitry Baturin and Lester Acosta of the University of Colorado Cancer Center Flow Cytometry Shared Resource for FACS, Trevor J. Isner and Amanda Stenzel for technical assistance with mouse and organoid experiments, and Thomas E. Finger, Katherine Fantauzzo, Heide Ford, Stephen Santoro, Santos Franco, Trevor J. Isner and Ian J. Purvis for helpful discussions and for critically reading the manuscript. Finally, we heartily thank Robin F. Krimm for sharing unpublished data. OTMSR is supported by the National Institutes of Health (P30DK116073). The University of Colorado Cancer Center is supported by the National Institutes of Health (P30CA046934).

### Competing interests

The authors declare no competing or financial interests.

### Author contributions

Conceptualization: C.M.P., E.T.L., P.J.D., J.v.M., L.A.B.; Data curation: C.M.P., J.K.S., A.S.H.; Formal analysis: C.M.P., C.E.W., A.S.H.; Funding acquisition: J.v.M., L.A.B.; Investigation: C.M.P., J.K.S., C.E.W., A.S.H., L.A.B.; Methodology: C.M.P., J.K.S., C.E.W., E.S., A.S.H., P.J.D.; Project administration: J.K.S., L.A.B.; Resources: H.I.L., P.J.D., J.v.M.; Software: E.S.; Supervision: L.A.B.; Validation: C.M.P., E.S.; Visualization: C.M.P., C.E.W.; Writing – original draft: C.M.P., J.K.S., C.E.W., H.I.L., E.S., E.T.L., P.J.D., J.v.M., L.A.B.; Writing – review & editing: C.M.P., J.K.S., C.E.W., H.I.L., E.S., E.T.L., P.J.D., J.v.M., L.A.B.

### Funding

This work was funded by the National Institutes of Health (T32GM141742 and F31DC020634-01 to C.M.P.; R01DC021865, R01DC018489, R01DC012383 and R21CA236480 to L.A.B.; R01AI167923 to J.v.M.). Open Access funding provided by University of Colorado. Deposited in PMC for immediate release.

### Data and resource availability

All relevant data and details of resources can be found within the article and its supplementary information.

### Peer review history

The peer review history is available online at https://journals.biologists.com/dev/lookup/doi/10.1242/dev.205259.reviewer-comments.pdf

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
