## [Peer Review File · Development (Cambridge, England)]

Tyrosine kinase inhibitors affect sweet taste and dysregulate fate selection of specific taste bud cell subtypes via KIT inhibition

Christina M. Piarowski, Jennifer K. Scott, Courtney E. Wilson, Heber I. Lara, Ernesto Salcedo, Andrew S. Han, Elaine T. Lam, Peter J. Dempsey, Jakob von Moltke and Linda A. Barlow

DOI: 10.1242/dev.205259

Editor: James Briscoe

Review timeline

Original submission:	19 September 2025
Editorial decision:	14 November 2025
First revision received:	10 February 2026
Accepted:	3 March 2026

Original submission

First decision letter

MS ID#: dev.205259

MS TITLE: Tyrosine kinase inhibitors affect sweet taste and dysregulate fate selection of specific taste bud cell subtypes via KIT inhibition

AUTHORS: Christina M. Piarowski; Jennifer K. Scott; Courtney E. Wilson; Heber I. Lara; Ernesto Salcedo; Andrew S. Han; Elaine T. Lam; Peter J. Dempsey; Jakob von Moltke; Linda A. Barlow

Dear Dr Barlow,

I have now received all the referees' reports on the above manuscript, and have reached a decision. The referees' comments are appended below.

As you will see, the referees express considerable interest in your work, but have some significant criticisms and recommend a substantial revision of your manuscript before we can consider publication. There appear to be three major issues that need addressing. First, as Reviewer 3 emphasises, there is "no direct evidence linking carbozantanib treatment to specific inhibition of c-Kit RTK activity in the sweet Type II TBC precursors," and Reviewer 3 suggests testing multiple TKIs to demonstrate that "the order of potency for decrease in sweet Type II TBC numbers correlates with their order of potency in inhibiting c-kit." Second, Reviewer 1 requests testing the converse experiment by "supplementing the medium with c-Kit ligand (SCF), which could potentially lead to an increase in sweet cells number and decrease in bitter/umami" to strengthen the organoid findings. Third, you should provide additional co-localization data: Reviewer 1 requests "double-label analysis (HCR ISH or IF) quantifying Tas1r2+/Kit+ vs Tas1r2+/Kit- cells," and Reviewer 2 suggests using T1r1 markers "to label umami TBCs for better understanding which subtypes of Gust+ CVP cells were increased."

If you are able to revise the manuscript along the lines suggested, which may involve further experiments, I will be happy to receive a revised version of the manuscript. Your revised paper will be re-reviewed by one or more of the original referees, and acceptance of your manuscript will depend on your addressing satisfactorily the reviewers' major concerns. Please also note that Development will normally permit only one round of major revision. If it would be helpful, you are

welcome to contact us to discuss your revision in greater detail. Please send us a point-by-point response indicating your plans for addressing the referees' comments, and we will look over this and provide further guidance.

Please attend to all of the reviewers' comments and ensure that you upload both a 'clean' version of your Word file, along with a highlighted version clearly showing where you have made changes in the revised manuscript. Please avoid using 'Tracked changes' in Word files as these are lost in PDF conversion. I should be grateful if you would also provide a point-by-point response detailing how you have dealt with the points raised by the reviewers in the 'Response to Reviewers' box. If you do not agree with any of their criticisms or suggestions please explain clearly why this is so.

Reviewer 1

While off-target inhibition of Receptor Tyrosine Kinases (RTKs) has long been suspected as a cause for the taste dysfunction (dysgeusia) reported by patients on TKI therapy, the underlying cellular mechanisms were previously unknown. This elegant work by Piarowski et al. suggests that the dysgeusia is not caused by traditional cytotoxic effects, such as inhibiting taste progenitor proliferation or survival. Instead, the authors propose a novel and more subtle mechanism: the TKIs alter the fate of differentiating taste cells through KIT inhibition, causing a shift that diminishes the sweet cell population while promoting a bitter/umami cell fate. The conclusion that cabozantinib acts via KIT inhibition is reinforced by the fact that conditional genetic ablation of Kit in the taste cell lineage recapitulated the cellular phenotypes observed with drug treatment. The study also nicely demonstrates that this loss of sweet cells was specific to the posterior tongue, offering new insights into the functional organization of the taste system.

A few suggestions include:

1. The organoid data presented in Figure 1 provide only limited support of the in vivo data. Although qPCR shows the downregulation of sweet cell markers, this pattern appears to correlate with the trend observed for Plcb2 expression (not statistically significant but following a similar trend), questioning the specificity of the effect. As the authors acknowledge, bitter/umami markers are overrepresented and do not change with the inhibition. This suggests that Kit signaling could be low already at the baseline culture condition. It might be worth testing the opposite approach, supplementing the medium with c-Kit ligand (SCF), which could potentially lead to an increase in sweet cells number and decrease in bitter/umami.
2. Data in CVP indicate Tas1r2 and Kit largely co-localize, and Kit-lineage cells are predominantly sweet, with only a small bitter/umami fraction. It is not clearly stated if there were any sweet cells that didn't express Kit. If such a cell population exists, could you add a supplemental double-label analysis (HCR ISH or IF) quantifying Tas1r2⁺/Kit⁺ vs Tas1r2⁺/Kit⁻ cells?
3. While the data are broadly consistent and support the proposed model, the relatively small number of mice (N = 3 per condition) with mixed genetic background could present some concerns. To illustrate variability and robustness of a phenotype without using additional animals, it would be helpful to include a supplemental figure based on the existing data from Figures 3 and 4, showing the per-bud counts of all Tas1r2⁺ and GUST⁺ cells.
4. Both the introduction and discussion state that cabozantinib inhibits KIT, but none of the cited references (Klaeger et al., 2017, Karaman et al., 2008, Davis et al., 2011) seem to show inhibition by this particular drug. Please include the citation that directly shows effect of cabozantinib on KIT receptor (e.g. Yakes et al., PMID: 21926191).

Reviewer 2

SUMMARY OF THE ADVANCE MADE IN THIS PAPER AND ITS POTENTIAL SIGNIFICANCE TO THE FIELD

This manuscript presents a compelling and well-executed study that identifies KIT as a regulator of taste bud cell subtype specification. Using a combination of lingual organoids, inducible genetic

models, and behavioral assays, the authors demonstrate that tyrosine kinase inhibitors (TKIs), particularly cabozantinib, selectively impair sweet taste by skewing the differentiation of type II taste bud cells. The work is conceptually novel, methodologically rigorous, and highly relevant to the field of developmental biology.

SUGGESTIONS TO AUTHORS

Major Comments

1. It would be helpful to have used the marker T1r1 to label umami TBCs for better understanding which subtypes of Gust+ CVP cells were increased in TKI treatment and Kit cKO.

Minor Comments

1. Ensure consistent use of terms such as "taste modality" vs "taste quality" to avoid confusion for readers outside of the taste field.

2. The manuscript uses terms like "sweet TBCs" and "GUST+ cells" consistently, but a concise table clarifying the subtype markers and their overlap across CVP and FFP would aid reader comprehension, especially for readers less familiar with taste cell subtype markers.

3. A brief mention of KIT's roles in other epithelial renewal systems (e.g., melanocytes, gut epithelium) could help contextualize its function in taste cell homeostasis.

Reviewer 3

SUMMARY OF THE ADVANCE MADE IN THIS PAPER AND ITS POTENTIAL SIGNIFICANCE TO THE FIELD

Given the important role of the sense of taste in quality of life, and the known detrimental effects of chemotherapeutic agents on gustation, the subject matter of this paper is of general interest. The paper provides information about a potential mechanism for the dysgeusia that results from treatment of cancer patients with tyrosine kinase-inhibiting (TKI) drugs: that this treatment results in a decrease in the abundance of sweet Type II receptors (TBCs) in taste buds. This change in TBC populations is argued to be due to inhibition of c-kit signaling, which is postulated to alter the differentiation of TBC precursor cells away from a sweet Type II receptor cell fate.

The data from organoid cultures and treatment of adult mice show that administration of carbozantinib, a TKI chemotherapeutic agent, results in a decreased number of sweet Type II TBCs (detected by the marker Tas1r2). Behavioral tests show a preference for sweet tastes in treated mice. A second category of experiments shows that conditional postnatal inactivation of Kit in the Pou2f3 (marker for type II TBC precursors) domain, results in fewer Tas1r2-positive TBCs in taste buds. Immunofluorescence for KIT shows that many Tas1r2-positive cells are also positive for KIT (but also, that there are many KIT+ cells in taste buds that are not Tas1r2-positive). Since conditional Kit inactivation results in a decrease in the number of Tas1r2-positive cells, and so does carbozantinib treatment, the authors reason that it is inhibition of the c-kit RTK pathway that must be the cause of a change in fate of the sweet Type II TBC precursors, leading to fewer sweet Type II TBCs in carbozantinib-treated mice.

The major issue with the paper is that there is no direct evidence linking carbozantinib treatment to specific inhibition of c-Kit RTK activity in the sweet Type II TBC precursors. Carbozantinib is documented to be a non-specific tyrosine kinase inhibitor, and so is likely to be acting on many RTKs in multiple cell types in the tongue. c-Kit itself interacts with multiple signaling pathways, so it is not clear that reducing its activity as a RTK is the dominant factor causing the TBC phenotype in taste buds (decrease in sweet Type II TBC numbers). Also, although c-Kit is expressed in sweet Type II TBCs, there are no data showing that the precursor cells of sweet Type II TBCs express c-Kit (something that might be expected for a cell-autonomous effect of c-Kit on cell fate).

SUGGESTIONS TO AUTHORS

An experiment that would go far toward establishing the connection between carbozantinib treatment and KIT inhibition, as the mechanism of decrease of sweet Type II TBCs, would be as follows: The investigators should identify multiple TKIs that have the effect (decrease of sweet Type II TBCs) in the organoid system, then show that the order of potency for decrease in sweet

Type II TBC numbers correlates with their order of potency in inhibiting c-kit. Although all drugs are "messy", this would argue strongly for a preferential effect of carbozantanib on c-kit.

First revision

Author response to reviewers' comments

Dear Dr. Briscoe,

Thank you for the opportunity to revise our manuscript in response to the 3 expert reviewers. We hope that we have satisfied their concerns and clarified and improved portions of the paper following their suggestions. We have also rebutted a small subset of requests, and these are detailed below. In particular, Reviewer 3's suggestion of an extensive set of tests of small molecule inhibitors of KIT is unlikely to yield interpretable results, as we explain below.

Response to reviewers

Reviewer 1

1. The organoid data presented in Figure 1 provide only limited support of the in vivo data. Although qPCR shows the downregulation of sweet cell markers, this pattern appears to correlate with the trend observed for Plcb2 expression (not statistically significant but following a similar trend), questioning the specificity of the effect. As the authors acknowledge, bitter/umami markers are overrepresented and do not change with the inhibition. This suggests that Kit signaling could be low already at the baseline culture condition. It might be worth testing the opposite approach, supplementing the medium with c-Kit ligand (SCF), which could potentially lead to an increase in sweet cells number and decrease in bitter/umami.

This is an excellent suggestion and an experiment we tried several times prior to initial submission of this paper. However, while small molecule drugs readily penetrate organoids, as evidenced by the reduction in sweet TBC markers with TKIs, we found that adding SCF to the culture medium did not increase *Tas1r2* nor reduce *Gnat3* (encodes Gustducin) expression. We suspect the tight basal epithelium at the organoid surface blocks penetration of peptides like SCF. If we were to develop inside-out organoid methodology, this could potentially improve SCF access to TBCs. However, establishing this protocol would have taken longer than the 3 months permitted for this revision and would not have been comparable to the organoid protocol used for TKI experiments. Finally, while it is true that the sweet cell phenotype in organoids is mild, it is also mild *in vivo*. Moreover, TKIs have no impact on any other aspect of TBC homeostasis that we measured in either organoids or mice, suggesting that the block to sweet TBC differentiation by these TKIs is specific.

2. Data in CVP indicate Tas1r2 and Kit largely co-localize, and Kit-lineage cells are predominantly sweet, with only a small bitter/umami fraction. It is not clearly stated if there were any sweet cells that didn't express Kit. If such a cell population exists, could you add a supplemental double-label analysis (HCR ISH or IF) quantifying Tas1r2⁺/Kit⁺ vs Tas1r2⁺/Kit⁻ cells?

This is an excellent point, and we now include additional quantification of the extent of *Kit* expression in both *Tas1r2*⁺ sweet TBCs and *GUST*⁺ bitter/umami TBCs in new panels in Figure S6. We also describe this information in the text. Briefly, *GUST*⁺ cells uniformly lack KIT expression, while most, but not all, *Tas1r2*⁺ sweet cells are *Kit*⁺.

3. While the data are broadly consistent and support the proposed model, the relatively small number of mice (N = 3 per condition) with mixed genetic background could present some concerns. To illustrate variability and robustness of a phenotype without using additional animals, it would be helpful to include a supplemental figure based on the existing data from Figures 3 and 4, showing the per-bud counts of all Tas1r2⁺ and GUST⁺ cells.

We agree that the numbers of mice and the numbers of new sweet cells in particular in each mouse are small, reflecting the slow genesis of this cell type in taste homeostasis. We also appreciate the opportunity to include the per-bud counts in new supplemental figures (Figures S7 and S8) that support the findings in Figures 3 and 4.

4. Both the introduction and discussion state that cabozantinib inhibits KIT, but none of the cited references (Klaeger et al., 2017, Karaman et al., 2008, Davis et al., 2011) seem to show inhibition by this particular drug. Please include the citation that directly shows effect of cabozantinib on KIT receptor (e.g. Yakes et al., PMID: 21926191).

The reviewer is correct in that the references cited are large screens of the spectrum of RTKs inhibited by a large panel of TKIs. Nonetheless, specific data are presented for cabozantinib, axitinib and sunitinib, including the IC50s and Kd's for VEGFRs and PDGFRs, as well as KIT. However, we appreciate the suggestion to reference citations where cabozantinib specifically inhibits KIT, and have included these in the text.

Reviewer 2:

SUMMARY OF THE ADVANCE MADE IN THIS PAPER AND ITS POTENTIAL SIGNIFICANCE TO THE FIELD

This manuscript presents a compelling and well-executed study that identifies KIT as a regulator of taste bud cell subtype specification. Using a combination of lingual organoids, inducible genetic models, and behavioral assays, the authors demonstrate that tyrosine kinase inhibitors (TKIs), particularly cabozantinib, selectively impair sweet taste by skewing the differentiation of type II taste bud cells. The work is conceptually novel, methodologically rigorous, and highly relevant to the field of developmental biology.

SUGGESTIONS TO AUTHORS

Major Comments

1. It would be helpful to have used the marker *T1r1* to label umami TBCs for better understanding which subtypes of *Gust+* CVP cells were increased in TKI treatment and *Kit* cKO.

A major limitation of the taste field is that the identity of umami sensing type II cells remains unsettled. While work from the Zuker group has put forth a model where each of the 5 taste modalities, sweet, sour, salt, bitter and umami, are transduced by separate TBCs, there are a lot of published data to suggest this is a simplified view, especially for umami. Work from Kim et al, 2003, which we cited previously, suggests that many *Tas1r2*⁺ cells, as well as many *GUST*⁺ cells, also express *Tas1r1* in both the CVP and FFP. Recent work from Lee et al., 2025 (<https://doi.org/10.1002/adv.202511309>) further supports this finding; a significant number of cells express both *Tas1r1* and *Tas1r2* and respond to both sweet and umami stimuli. Further, in another project we are writing up we find that *Tas1r1* is indeed expressed in sweet cells and in a subset of bitter cells. Thus, umami cells may not represent an independent cell population. Any change in *Tas1r1* expression is therefore unlikely to be informative, as *Tas1r2*⁺/*Tas1r1*⁺ cells would be lost while *GUST*⁺/*Tas1r1*⁺ cells may be increased. Lastly, TAS1R1 forms a heterodimer with TAS1R3 to detect umami tastants. *Tas1r1* is expressed in many *Tas1r3*-negative cells (Kim et al. 2003), suggesting that *Tas1r1* expression alone cannot determine sensitivity to umami.

We have added a synopsis of this explanation, limited to published findings, to the “Limitations” section of the manuscript, and hope this satisfies the reviewer’s concern.

Minor Comments

1. Ensure consistent use of terms such as “taste modality” vs “taste quality” to avoid confusion for readers outside of the taste field.

We have revised to “modality” throughout.

2. The manuscript uses terms like "sweet TBCs" and "GUST+ cells" consistently, but a concise table clarifying the subtype markers and their overlap across CVP and FFP would aid reader comprehension, especially for readers less familiar with taste cell subtype markers.

This information is now presented in tabular form (Table 1).

3. A brief mention of KIT's roles in other epithelial renewal systems (e.g., melanocytes, gut epithelium) could help contextualize its function in taste cell homeostasis.

We have expanded discussion of KIT's role in other renewing epithelial systems, such as melanocytes and spermatogonia, as well as discussion of potential downstream signaling targets. Additionally, we have included discussion of KIT's role during infection and injury in other epithelial systems, including within the intestinal epithelium. We specifically discuss a complementary paper debuting the *Pou2F3^{CreER/+};Kit^{fl/fl}* conditional knockout line authored by 2 of our co-authors to investigate KIT signaling within the intestinal epithelium during helminth infection. KIT function regulates the hyperplasia of intestinal tuft cells, which, like type II TBCs, require Pou2F3 for their generation. We hope these additional discussions further contextualize KIT's role within the taste epithelium.

Reviewer 3: SUMMARY OF THE ADVANCE MADE IN THIS PAPER AND ITS POTENTIAL SIGNIFICANCE TO THE FIELD

Given the important role of the sense of taste in quality of life, and the known detrimental effects of chemotherapeutic agents on gustation, the subject matter of this paper is of general interest. The paper provides information about a potential mechanism for the dysgeusia that results from treatment of cancer patients with tyrosine kinase-inhibiting (TKI) drugs: that this treatment results in a decrease in the abundance of sweet Type II receptors (TBCs) in taste buds. This change in TBC populations is argued to be due to inhibition of c-kit signaling, which is postulated to alter the differentiation TBC precursor cells away from a sweet Type II receptor cell fate.

*The data from organoid cultures and treatment of adult mice show that administration of **carbozantinib**, a TKI chemotherapeutic agent, results in a decreased number of sweet Type II TBCs (detected by the marker *Tas1r2*). **Behavioral tests show a preference for sweet tastes in treated mice.***

Please note the drug is **cabozantinib**, and that drug-treated mice **have a blunted behavioral response** to sweet in both 2-bottle preference and limited access lickometer tests.

*A second category of experiments shows that conditional postnatal inactivation of *Kit* in the *Pou2F3* (marker for type II TBC precursors) domain, results in fewer *Tas1r2*-positive TBCs in taste buds. Immunofluorescence for KIT shows that many *Tas1r2*-positive cells are also positive for KIT (but also, that there are many KIT+ cells in taste buds that are not *Tas1r2*-positive).*

In response to Reviewer 1, we have quantified *Tas1r2/Kit* and GUST/KIT co-expression and find that most, but not all, sweet cells are *Kit*⁺, while all *Kit*⁺ cells are sweet cells. We also show that GUST⁺ bitter/umami cells are all KIT^{neg}, in new panels in Supplemental Figure 6.

*Since conditional *Kit* inactivation results in a decrease in the number of *Tas1r2*-positive cells, and so does carbozantinib treatment, the authors reason that it is inhibition of the c-kit RTK pathway that must be the cause of a change in fate of the sweet Type II TBC precursors, leading to fewer sweet Type II TBCs in carbozantinib-treated mice.*

The major issue with the paper is that there is no direct evidence linking carbozantinib treatment to specific inhibition of c-Kit RTK activity in the sweet Type II TBC precursors. Carbozantinib is documented to be a non-specific tyrosine kinase inhibitor, and so is likely to be acting on many RTKs in multiple cell types in the tongue. c-Kit itself interacts with multiple signaling pathways, so it is not clear that reducing its activity as a RTK is the dominant factor causing the TBC

phenotype in taste buds (decrease in sweet Type II TBC numbers). Also, although c-Kit is expressed in sweet Type II TBCs, there are no data showing that the precursor cells of sweet Type II TBCs express c-Kit (something that might be expected for a cell-autonomous effect of c-Kit on cell fate).

We understand the reviewer's perspective but believe that our interpretation of our data as a whole is sound. Namely that sweet type II TBC differentiation depends on KIT function, and infer that KIT is required in type II precursor cells that give rise to sweet or bitter/umami cells. The data that support our model are as follows.

1. In cabozantinib-treated mice, sweet cells are lost and bitter/umami cells increased. We show this switch in cell fate is due to the impact of the drug on newly differentiated cells and not to effects on sweet cell survival or transdifferentiation of sweet cells into bitter/umami cells.
2. Cabozantinib is well documented to inhibit KIT at nM concentration, and is used to treat prostate cancer patients to target KIT, where KIT activation is causal.
3. We show that *KitCreER;YFP* lineage tracing results predominantly YFP⁺ sweet cells, but some bitter/umami cells are also labeled; however, we now include additional data that bitter/umami cells lack KIT expression, thus the YFP label must be "inherited" from a type II precursor cell.
4. POU2F3 is uniquely expressed in type II cells and in some taste precursor cells (Matsumoto et al., Nat Neurosci). Deletion of *Kit* in *Pou2f3CreER* mice phenocopies cabozantinib treatment, i.e., sweet cells are lost and bitter/umami cells are gained.
5. Thus, we reason that cabozantinib treatment likely inhibits KIT in the type II cell lineage in taste epithelium.

SUGGESTIONS TO AUTHORS

An experiment that would go far toward establishing the connection between cabozantinib treatment and KIT inhibition, as the mechanism of decrease of sweet Type II TBCs, would be as follows: The Investigators should identify multiple TKIs that have the effect (decrease of sweet Type II TBCs) in the organoid system, then show that the order of potency for decrease in sweet Type II TBC numbers correlates with their order of potency in inhibiting c-kit. Although all drugs are "messy", this would argue strongly for a preferential effect of cabozantinib on c-kit.

This is an interesting line of inquiry, but unlikely to yield interpretable results.

TKIs bind and competitively block the ATP binding pocket of RTKs. As many RTKs, such as VEGFRs, PDGFRs, KIT, etc., have structurally similar binding pockets and all TKIs bind the pocket albeit with different affinities; thus, anti-angiogenic TKIs are not specific. Consequently, there are dozens of TKIs known to inhibit KIT at nM concentrations, but to date there are no TKIs that specifically, and uniquely, inhibit KIT. These are observations that we have made via perusal of a large literature and information we have assembled from published reports and vendor sites, e.g., SelleckChem. As none of these drugs is selective for KIT, by extension they also inhibit other RTKs, many we know to be expressed in a spectrum of cell types in taste epithelium. Thus, with changes in potency of KIT inhibition, other RTKs would be impacted, which would likely affect other processes in taste homeostasis. We do not think this approach would be fruitful in addressing the reviewer's concern. Rather, the fact that conditional deletion of *Kit* phenocopies the impact of cabozantinib strongly supports our conclusion that the effect of the drug is via KIT inhibition.

Second decision letter

MS ID#: dev.205259R1

MS TITLE: Tyrosine kinase inhibitors affect sweet taste and dysregulate fate selection of specific taste bud cell subtypes via KIT inhibition

AUTHORS: Christina M. Piarowski; Jennifer K. Scott; Courtney E. Wilson; Heber I. Lara; Ernesto Salcedo; Andrew S. Han; Elaine T. Lam; Peter J. Dempsey; Jakob von Moltke; Linda A. Barlow
ARTICLE TYPE: Research Report

Dear Dr Barlow,

I am happy to tell you that your manuscript has been accepted for publication in Development, pending our standard publication integrity checks.

Reviewer 1

The authors have nicely addressed my suggestions.

Reviewer 2

SUMMARY OF THE ADVANCE MADE IN THIS PAPER AND ITS POTENTIAL SIGNIFICANCE TO THE FIELD

The same as originally provided.

SUGGESTIONS TO AUTHORS

The authors have addressed the concerns in the revised version.

Reviewer 3

SUMMARY OF THE ADVANCE MADE IN THIS PAPER AND ITS POTENTIAL SIGNIFICANCE TO THE FIELD

Given the important role of the sense of taste in quality of life, and the known detrimental effects of chemotherapeutic agents on gustation, the subject matter of this paper is of general interest. The paper provides evidence that a mechanism for the dysgeusia that results from treatment of cancer patients with tyrosine kinase-inhibiting (TKI) drugs, specifically cabozantinib (which is used in the treatment of prostate cancer), results in a decrease in the abundance of sweet Type II receptors (TBCs) in taste buds. The authors present evidence that the change in TBC populations is due to inhibition of c-kit signaling, which is postulated to alter the differentiation of TBC precursor cells away from a sweet Type II receptor cell fate.

SUGGESTIONS TO AUTHORS

The authors point out the misspelling of cabozantinib in the prior review; thank you.

Although it would have been preferable to this reviewer to see a pharmacological experiment addressing of the specificity/potency of cabozantinib action on sweet Type II TBC numbers in organoids, as suggested, the authors disagree about the importance/interpretability of such experiments.

Instead, the authors argue that several lines of evidence are consistent with their conclusion that the effect of cabozantinib is via KIT inhibition. They have also provided new data to deal with noted issues concerning cell type numbers.

The revised version of the paper is acceptable for publication.